

# Hydrogen isotope fractionation affects the identification and quantification of tree water sources in a riparian forest

Adrià Barbeta[1], Sam P. Jones[1], Laura Clavé[1], Lisa Wingate[1], Teresa E. Gimeno[1,2,3], Bastien Fréjaville[1], Steve Wohl[1], Jérôme Ogée[1]

[1]INRA, UMR ISPA, F-33140, Villenave d'Ornon, France
[2]BC3 - Basque Centre for Climate Change - Klima Aldaketa Ikergai, E-48940, Leioa, Spain
[3]IKERBASQUE, Basque Foundation for Science, 48008 Bilbao, Spain

*Correspondence to*: Adrià Barbeta (adria.barbeta-margarit@inra.fr)

**Abstract.** We investigated plant-water sources of an emblematic refugial population of *Fagus sylvatica* (L.) in the Ciron river gorges in South-Western France using stable isotopes. The stable isotopes of water are a powerful tracer of water fluxes in the soil-plant-atmosphere continuum. It is generally assumed that no isotopic fractionation occurs during root water uptake, and that xylem water isotopes effectively reflect source water isotopes. However, recent studies showed that under certain conditions the isotopes in plant water do not reflect any of the potential sources considered. Highly resolved datasets covering a range of environmental conditions could shed light on possible plant-soil fractionations processes. In this syudy, the hydrogen ($\delta^2$H) and oxygen ($\delta^{18}$O) isotope compositions of all potential tree water sources and xylem water were measured bi-weekly over an entire growing season. Using Bayesian isotope mixing models (MixSIAR), we then quantified the contribution of the considered sources to xylem water of *F. sylvatica* and *Quercus robur* (L.) trees. Based on $\delta^{18}$O data alone, both species used a mix of top and deep soil water over the season, with *Q. robur* using soil water relatively deeper than *F. sylvatica*. The contribution of stream water appeared to be marginal despite the proximity of the trees to the stream, as already reported for other riparian forests. Xylem water $\delta^{18}$O could always be interpreted as a mixture of deep and shallow soil waters, but the $\delta^2$H of xylem water was often more depleted than any other possible water source. We argue that an isotopic fractionation in the unsaturated zone and/or within the plant tissues could underlie this unexpected $\delta^2$H depletion of xylem water, as already observed in halophytic and xerophytic species. By means of a sensitivity analysis, we found that the estimation of plant-water sources using isotope mixing models was largely affected by this isotopic $\delta^2$H depletion. A better understanding of what causes this isotopic separation between xylem and source water is urgently needed.

## 1 Introduction

### 1.1 Why an improved understanding of tree water use is needed?

The combination of altered precipitation regimes and increasing temperatures is already affecting terrestrial ecosystem function profoundly and promoting changes in land cover (Allen et al., 2015). As land-atmosphere water fluxes are largely regulated by the biosphere, especially forested areas (Good et al., 2017), future changes in energy, carbon and water exchanges between





ecosystems and the atmosphere will also be influenced by altered transpiration dynamics in response to variations in soil
moisture (Berg et al., 2016; Seneviratne et al., 2013). In this context, it is a priority to better understand how spatial and
temporal dynamics of the soil-plant-atmosphere water continuum will be affected. In particular, an enhanced knowledge of
belowground processes will not only reduce uncertainty in the projections of biosphere-atmosphere feedbacks, but will also
improve ecohydrology models (Cáceres et al., 2015). Soil-plant interactions and species-specific water use can also provoke
shifts in forest species' distributions through readjustments in species abundance (Clark et al., 2015), such mechanisms may
help understand how climate refugia can be buffered during climate change and facilitate the persistence of important
biodiversity hotspots (McLaughlin et al., 2017).

## 1.2 The isotopic tracing method to study tree water use

Studying belowground processes is technically challenging because their direct observation usually requires destructive,
expensive and/or time-consuming approaches that hamper the assessment of temporal and spatial variability within catchments
(Fahey et al., 2017). Water isotope tracing has become the commonly used methodology to study plant water uptake (e.g.
Dawson et al. 2002, see also references below) and is based on two main principles. Firstly (H1), it is assumed that isotopic
fractionation during root water uptake does not occur (Allison et al., 1984; Dawson and Ehleringer, 1991; Ehleringer and
Dawson, 1992; White et al., 1985). However, some exceptions have challenged this assumption most notably in halophytic
and xerophytic plants for the moment (Ellsworth and Williams, 2007; Lin and Sternberg, 1993). Secondly (H2), it is essential
that all potential water sources are identified and accessible, and their isotopic compositions differ enough to distinguish their
contribution to xylem water (Ehleringer & Dawson, 1992 and references therein). However, plants have access to various
pools of water with contrasting isotopic compositions. Processes underlying the variability in source water isotopic
composition include, the temporal variability in rainfall water isotopes and the soil depth reached by infiltrating water (Allison
and Hughes, 1983; Brooks et al., 2010), the evaporative enrichment of water in surface soil layers (Allison, 1982; Sprenger et
al., 2016), the seasonality of groundwater and rock moisture recharge (Oshun et al., 2015) or the relatively higher concentration
of heavy isotopes in fog compared to rain water (Scholl et al., 2011). In the last decade, the development of laser-based isotopic
analysers have increased the throughput of water isotope measurements, providing opportunities to carry out observational
studies at higher temporal and spatial resolution for both deuterium and oxygen isotope values systematically (Martín-Gómez
et al., 2015).

## 1.3 Possible caveats of the isotopic tracing method

This conceptual framework has succeeded in advancing our understanding of plant water uptake (Ehleringer and Dawson
1992; Dawson et al. 2002). However, it occasionally leads to rather unexpected results. For instance, a pioneering study using
hydrogen isotopes alone concluded that mature streamside riparian trees in a semi-arid catchment did not use stream water but
were dependent on an unidentified, relatively more depleted, water source, hypothesized to reflect groundwater (Dawson and
Ehleringer, 1991). On the other hand, small trees seemed to rely on stream water. Another study conducted in a seasonally dry



conifer forest in Oregon found that the $\delta^{18}O$ and $\delta^2H$ of soil and tree water were similar during the dry season, and clearly distinct from stream water, even when sampled near the stream (Brooks et al. 2010). This led to the two-water-worlds (TWW) hypothesis whereby the first depleted rainfall water after a rainless summer becomes locked into small soil pores no longer
participating in river flow nor mixing with subsequent rain events. This water pool eventually participates in soil evaporation but mainly remains in the soil until being used by plant transpiration during the following dry summer. In light of this TWW hypothesis, Bowling *et al.* (2017) revisited the study site of Dawson and Ehleringer (1991) many years later and re-measured hydrogen but also oxygen water isotopes in xylem water, soil water at different depths and stream water, in addition to groundwater and snowmelt water this time. They argued that, if the TWW hypothesis was true, the soil, still dry after winter,
would have been recharged in spring during snow melt, leading to more depleted snowpack water being locked in small soil pores and being used by the trees in summer only. Although the vertical distribution of soil water isotopes following snowmelt seemed consistent with the TWW hypothesis, neither snow melt water, nor groundwater could be identified as an alternative source to stream water. They concluded that soil moisture seemed to be the most likely water source for these riparian trees, but that the dual isotope approach could not unambiguously determine the water sources of these riparian trees.

**1.4 Rock moisture as an alternative plant water source?**

Plant water source studies in which xylem water isotopes do not match any of the considered sources often acknowledge that a relevant source of water may not have been sampled (Bowling et al., 2017; Geris et al., 2017), thus implying that hypothesis (H2) was not met. Not many studies sample rock moisture, but it has been shown that water stored in rocky layers can contribute to plant transpiration, sometimes more than saturated soil layers (Barbeta & Peñuelas, 2017 and refs therein). Indeed, rock
moisture can represent up to 27% of annual rainfall, and can be taken up by trees during the dry season (Rempe and Dietrich, 2018). Moreover, the water stored in soil rock fragments can carry an isotopic signal distinct to that of soil water or groundwater, either more depleted (in the case of $\delta^2H$ in Oshun *et al.*, 2015), or more enriched (Palacio et al., 2014; Rong et al., 2011). Such variable and contrasted isotopic effects of lithology are to be expected for differing minerals, and can even cause different fractionations for the hydrogen and oxygen isotopes (Meißner et al., 2013; Oerter et al., 2014). Thus, wherever
weathered rocks constitute a large fraction of the soil volume, the isotopic composition of rock water should be measured as it could constitute a relevant, alternative plant water source.

**1.5 Evidence of isotope fractionation during root water uptake**

Although we cannot rule out whether rock water in the carbonate-rich soil of Bowling et al. (2017) was a significant source of water for trees or caused any unexpected isotopic effects, the very clayey soil texture reported by Brooks et al. (2010) seems
less likely to contain a large rock water component. Furthermore, if the water contained in small pores does not interact with the more mobile water (the TWW hypothesis), in summer, when only small pore water is accessible to the trees, there should be an isotopic match between soil pore and xylem water, unless isotopic fractionation occurs during root water uptake. In this context, a recent controlled experiment conducted on potted avocado (*Persea americana*) trees has revealed isotopic



fractionation during root water uptake (Vargas et al. 2017), questioning the assumption (H1) that fractionation only occurs in

halophytic and xerophytic plants. Vargas et al. (2017) showed that *P. americana* plants discriminated against hydrogen isotopes about 10 times more than oxygen isotopes during water uptake, and this discrimination increased with soil water loss, porosity and particle size. Interestingly, the datasets reported by Brooks et al. (2010) and Bowling *et al.* (2017) contain a substantial number of xylem water samples that occupy the $\delta^{18}O$-$\delta^2H$ space well below the soil water line, suggestive of deuterium fractionation processes during root water uptake. In fact, a growing number of studies are observing depleted xylem

water isotopes compared to considered sources (Brooks et al. 2010; Bowling *et al.*, 2017; Evaristo *et al.*, 2017; Geris *et al.*, 2017; Wang *et al.*, 2017; De Deurwaerder *et al.*, 2018), suggesting that isotopic fractionation during root water uptake may be more common than previously thought. If such fractionation processes are not taken into account, the estimation of plant water sources may be miscalculated. The effect of deuterium fractionation on plant source water quantification was addressed in a recent study that concluded erroneous results could be obtained if one used a simple mass balance approach using deuterium

isotopes only (Evaristo et al. 2017). On the other hand, it was also concluded that results were less sensitive to deuterium fractionation when both deuterium and oxygen isotopes were combined within a Bayesian inference approach (Evaristo et al. 2017).

The aim of this study was to identify the water sources of a refugial population of *Fagus sylvatica* (L.) in SW France, nearing the southernmost distribution limit of this species. Evidence from studies of population genetics (De Lafontaine et al.,

2013) and *in situ* soil macrofossil charcoals radiocarbon-dated back to more than 40 kyr before present, when the area was a periglacial desert (de Lafontaine *et al.*, 2014) indicate that the Ciron valley acted as a climate refugia during the Last Glacial Maximum (de Lafontaine et al., 2014; Timbal and Ducousso, 2010). Thereafter *F. sylvatica* expanded northwards and colonized the areas of its current distribution range from this and other populations in Southern Europe  (Gavin et al., 2014). The population is hypothesised to have persisted there since the Last Glacial Maximum (de Lafontaine *et al.*, 2014) because

of an array of edaphic, thermal and hydric features decoupled from the surrounding regional environment. These features include: convergent topography, frequent fog, short distances to a stream and complex lithology. We thus sampled stable isotopes and applied isotope mixing models to the datasets to quantify the relative contribution of different water sources to both *F. sylvatica* and the more regionally widespread *Quercus robur* (L.), in an attempt to understand better the ecohydrological mechanisms shaping this refugium. To do this we adapted our methodology to address a number of the caveats raised above

including the sampling of all potential water sources, including fog and rock water, in addition to measuring both water isotopes to identify better possible isotopic fractionation during root water uptake. We used a Bayesian inference approach to quantify how plant source water varied seasonally, between species and with distance to the river. In parallel to the ecological focus of our study, we analysed an extensive isotope dataset for evidence of isotopic fractionation during root water uptake by mature trees in the field and its potential effect on the quantification of tree water sources.




## 2. Methods

### 2.1 Study site and experimental design

European beech (*Fagus sylvatica* L.) is a deciduous broadleaved tree species distributed across most of Western and Central Europe. The population that is the focus of this study is found along a mixed riparian forest on the karstic canyon formed by the Ciron, a tributary of the Garonne river, in Gironde, a south-western French region (44°23 N, 0°18 W, 60 m a.s.l.). The soil at the studied site has a fine texture and is slightly less organic, than the sandy soils found in the surroundings and typical of the Aquitaine basin (Table S1). Importantly, the presence of limestone rocks weathered to various degrees creates a

distinguishable carbonate-rich C horizon between 50 and 120 cm belowground (Table S1). Interestingly this European beech population is restricted either to the sheltered Ciron ravine or to slightly more distant sites (100m) located on irregular microtopography with small karstic depressions. In this riparian forest *F. sylvatica* trees coexist with other deciduous species such as *Quercus robur* L., a regionally common tree species that dominates the canopy further away from the river. Other tree species within the riparian forest are *Carpinus betulus* L., *Alnus glutinosa* L., *Corylus avellana* L. and *Tilia platyphyllos* Scop..

At the riparian forest limits beyond the chalky soil areas plantations of *Pinus pinaster* Ait. have been clear cut. The studied area has a mean annual temperature of 12.9°C and a mean annual precipitation of 812.9 mm evenly distributed along the year (data for the period 1897-2015). Since 1897, mean annual temperature increased by +1.0°C ($P < 0.001$), whereas precipitation did not show any trend.

Early in 2017, three field plots with different conditions were set up within the riparian forest. Two of the plots were

located on opposite sides of the river (NE and SW) to explore exposition effects and a third plot was instrumented, adjacent to an open area formerly covered by a pine plantation, chosen to explore the effect of forest fragmentation on the microclimate (notably fog occurrence). In each of the plots, we selected five mature *F. sylvatica* and three *Q. robur* individuals of 80-150 years and all occupying dominant positions in the canopy. In addition, we selected six non-dominant *F. sylvatica* trees in each of the two plots to explore the effect of tree size (Dawson & Ehleringer 1991, 1993). All selected trees were sampled bi-weekly

from mid-April to early November 2017. In order to identify the xylem water isotopic signal, several twigs were collected from every tree, rapidly peeled to remove bark and phloem then placed in an air-tight Exetainer® sealed with Parafilm® and kept in a cool box until they were stored in the lab at 4°C. For four trees (three *F. sylvatica* and one *Q. robur*), the canopy could not be accessed so we extracted xylem samples from coarse roots with an increment borer. Three soil cores per plot, located randomly amongst the sampled trees, were extracted with a soil auger. Each soil core was split into top soil (0-10 cm)

and deep soil (from 70-80 to 110-120 cm depending on the depth of the rocky layer). Soil samples were placed in 20 mL vials with positive insert screw-top caps, sealed with Parafilm® and kept in a cool box until they were stored in the lab at 4°C. From July onwards, we also sampled limestone rocks. We dug horizontally into rocky edges to avoid the effect of evaporation, and collected one sample per plot and sampling date.

In addition to soil, xylem and rocks, we collected for every date water from the stream, groundwater from a well

located *ca*. 50 m from the river, and fog and rain water from collectors installed in a small open area about 100 m away from



one of our plots. Both rain and fog collectors were connected *via* a funnel to a thermally insulated water reservoir buried in the ground with minimal contact with the open air following the recommendations of the GNIP network (http://www-naweb.iaea.org/napc/ih/IHS_resources_gnip.html). The fog collector was custom-built following the design of the single-stage Caltech Active Strand Cloud water Collector (CASCC2, Demoz et al. 1996). This design has been shown to be well suited for

water isotope studies (Spiegel et al. 2012). According to the theory presented in Demoz et al. (1996) our one-stage fog collector is ill-designed to collect small fog events (i.e. clouds with droplet sizes of 7 µm or less, corresponding to a liquid water content of less than 0.01 g m$^{-3}$) but such fog events are unlikely to have any significant contribution to the water source of the trees (i.e. less than 0.5 L h$^{-1}$ tree$^{-1}$, assuming a surface of exchange of 60 m$^2$ tree$^{-1}$ and an average wind speed through the tree crown of 0.2 m s$^{-1}$ during such fog events). On the other hand, any fog event with enough water made up of droplets larger than 7µm

in diameter was (in theory) collected, and the isotopic composition of this fraction of the cloud was expected to be representative of the entire cloud, because the isotopic composition of cloud droplets is independent of their size (Spiegel et al. 2012).

          Daily meteorological data was taken from a weather station located at about 20 km from the studied site. Long-term (1897-present) monthly temperature and precipitation data was also available from another weather station located 16 km away

from the studied area. Streamflow data was obtained from a stream gauge located about 4 km downstream of our study area.

## 2.2 Water extraction and determination of stable isotope composition

The water contained in soil, xylem and rock samples was extracted using a cryogenic vacuum distillation system based on the design and methodology described by Orlowski *et al.*, (2013). A detailed description of the system used is available in Jones

*et al.* (2017). Briefly, the pressure in the "vacuum" line was set at less than 1 Pa at the start of the extraction (i.e. when the samples were still frozen in liquid nitrogen). The samples were then gradually (within 1h) heated up to 80°C (soils) or 60°C (xylem) for 2.5 hours. The pressure line was continuously recorded using sub-atmospheric pressure sensors (APG100 Active Pirani Vacuum Gauges, Edwards, Burgess Hill, UK) to check that the lines remained leak-tight throughout the entire extraction. Gravimetric water content was assessed for each sample using the sample weight before and after water extraction. We also

checked that the water extraction had been completed by oven drying all samples at 105°C for 24h and re-weighing them.

          The isotopic compositions ($\delta^2$H and $\delta^{18}$O) of the different waters were measured with an off-axis integrated cavity optical spectrometer (TIWA-45EP, Los Gatos Research, USA) coupled to an auto-sampler. Details on the processing and post-correction of water samples can be found in Jones *et al.*, (2017). The presence of organic compounds (ethanol, methanol and/or other biogenic volatile compounds) in water samples can lead to large isotopic discrepancies in laser-based analyses (Martín-

Gómez *et al.*, 2015; Wassenaar et al. 2018). Organic compounds are found in certain soil types (Orlowski et al., 2018), but are more typically found in water extracted from plant tissues (Zhao et al. 2011). Post-corrections to account for the presence of organic compounds in water can be applied, based on metrics from the measured absorption spectrum (Brian Leen et al., 2012; Schultz et al., 2011). Nonetheless, these post-processing functions must be developed for each individual instrument. Thus





following Schultz *et al.* (2011), we developed our own post-corrections by analysing milliQ waters mixed with methanol
and/or ethanol at various concentrations and by fitting the measured deviation from the expected isotope ratio to the narrow-
and broad-band metrics provided by the instrument. We verified the performance of our correction with the contaminated
standard WICO5 (Wassenaar et al. 2018). Xylem water samples generally exhibited higher narrow and broad band metrics
compared to rain or even soil water samples but the corrections on xylem samples were always quite small (ca. 1.5 ‰ for $\delta^2$H
and 0.7 ‰ for $\delta^{18}$O) compared to the correction we had to apply on the WICO5 sample for some of our water-alcohol mixtures.

## 2.3 Data analysis

The relationships between xylem water and its potential sources were compared at the plot level. All the analyses described
below are also calculated at the plot level. Because there was no significant difference found between the studied plots, "plot"
was set as a random factor. To assess whether there was a mismatch between tree xylem water and its potential sources, the
concept of the line-conditioned excess proposed by Landwehr & Coplen (2006) was used: LC-excess = $\delta^2$H – $a$ $\delta^{18}$O - $b$, where
$a$ and $b$ correspond to the slope and intercept of the Local Meteoric Water Line (LMWL), respectively. However, because the
source water for a tree is more likely to be made of soil water than rain water directly, we modified the equation above and
computed the deviation of a given xylem water with respect to the soil water line from the same plot and date:

SW-excess = $\delta^2$H – $\sigma$ $\delta^{18}$O - $\Lambda$,

where $\sigma$ and $\Lambda$ are the slope and intercept of the soil water line for a given plot and date, respectively, and $\delta^2$H and $\delta^{18}$O
correspond to the isotopic composition of a xylem water sample collected on that plot at that date. The slope and intercept
$\sigma$ and $\Lambda$ were computed by performing a linear regression on all the soil water isotope data from the surface and deep horizons
collected at a given plot and date. The SW-excess of xylem water is an indicator of the $\delta^2$H separation between xylem samples
and their corresponding soil water lines. Positive SW-excess values indicate xylem samples that are more enriched in deuterium
than the soil water line (and are thus positioned above soil water in a $\delta^{18}$O-$\delta^2$H diagram), while negative SW-excess values
indicate xylem samples that are more depleted in deuterium than the soil water line (and are thus positioned below soil water
in a $\delta^{18}$O-$\delta^2$H diagram).

The contribution of different water sources to that of xylem water was estimated using the isotope mixing models of
the *MixSIAR* package (Stock and Semmens, 2016) in *R* (R Core Development Team, 2012). Models were ran in the script
version of the package, and the number of Markov chain Monte-Carlo iterations was increased until convergence was reached
and the results for the Gelman and Geweke diagnostics were acceptable. We grouped together trees of the same plot, species
and date altogether, and thus specified the residual error term in the isotope mixing models (Parnell et al., 2010). The potential
tree water sources that we considered were restricted to the top and deep soil water and stream/groundwater. Stream and ground
waters were pooled together as they were isotopically indistinguishable. Fog and rock moisture were not included as potential
water sources with this approach because their isotopic signatures were very distant to xylem waters in a $\delta^{18}$O-$\delta^2$H diagram,



and because there were only a limited number of campaigns measured. In order to test the sensitivity of *MixSIAR* to different data inputs, the models were run with four different input data; (1) $\delta^2H$ and $\delta^{18}O$, (2) $\delta^2H$ and $\delta^{18}O$ after subtracting the SW-excess from the $\delta^2H$ of xylem samples, (3) only $\delta^{18}O$ and (4) only $\delta^2H$.

The spatial, temporal, species-specific and size-related statistical comparisons between the isotopic compositions of
grouped samples were analyzed using the package *lme4* in *R*, where plot and date were used as random factors where necessary. In order to understand the factors driving the observed SW-excess of xylem water, we fitted Generalized Linear Mixed Models (GLMM) including soil moisture (top and deep), soil water isotopes, rock moisture isotopes, tree diameter (DBH), as well as rainfall and vapor pressure deficit (VPD) prior to sampling and using the tree species as an explanatory variable. We selected the best model by means of the second-order Akaike Information Criteria (AIC). Given the water source contributions
estimated with *MixSIAR* using different input data, we assessed their correlations with top and deep soil moisture, rainfall and VPD, also using GLMM from *lme4*.

## 3. Results

### 3.1 Environmental conditions

The mean temperature of the 2017 (April-November) growing season was 0.4°C warmer than the long-term average, but 0.5°C cooler than the average of the last 25 years. Precipitation during the 2017 growing season was 20% lower than the long-term average but close to the average of the last 25 years. There was a clear deficit in precipitation from the previous winter (estimated from December 2016 to March 2017) that caused a 43% reduction in streamflow compared to the 2000-2017 average, throughout the entire growing season (Fig. 1). Deep soil layers progressively dried over the entire growing season up
to the last sampling campaign in November, while top soil moisture was usually higher but also more variable, with relatively high levels at the beginning and end of the season and levels as low as the deep soil layer only in mid-summer (Fig. 1). Based on the water retention properties of top and deep soil layers, we estimated that the permanent wilting point was reached in the top soil only in early September, and from late July to the end of the season in the deep soil. On average, the volumetric water content of limestone rocks was around 11.75%, which is comparable to that of deep soil. We did not determine the water
potential that was required to extract this rock water but we expect it to be lower than the extracted deep soil waters.

### 3.2. Stable isotopes of tree water sources

The local meteoric water line (LMWL) was constructed with rain water isotope data collected monthly since February 2007 at a Global Network of Isotopes in Precipitation (GNIP) station located in Cestas, France (Fig. 2). The rain data collected bi-
weekly plotted closely to this line and ranged, from -7.0‰ to -1.3‰ in $\delta^{18}O$ and -46.8‰ to -5.4‰ in $\delta^2H$ on the VSMOW-SLAP scale (Fig. 2). Fog water ranged from between -6.5‰ and -0.9‰ in $\delta^{18}O$ and between -32.4‰ and -8.4‰ in $\delta^2H$ and



was not significantly different from rain water (Fig. S1). Stream and groundwater had isotopic compositions that were not statistically different and very stable over time (-5.9 ± 0.2‰ in $\delta^{18}O$ and -36.8 ± 0.8‰ in $\delta^2H$).

Soil water samples occupied the $\delta^2H$-$\delta^{18}O$ space on the right side of the LMWL (Fig. 2). On all plots, top soil water

was significantly more enriched than deep soil ($P < 0.001$ for both isotopes) as a result of water evaporative enrichment at the soil surface. The resultant soil water line (SWL) had a mean slope of 5.17 (ranging from 4.01 to 9.99), that is significantly smaller than the slope of the LMWL (6.73). The difference in $\delta^{18}O$ between top and deep soil water was significantly smaller in the plot within a mixed broadleaved forest ($P < 0.05$), suggesting that soil evaporation was probably lower at this plot.

Over the season, rainfall amounts over the 15 days preceding each sampling campaign had a negative effect on top

soil water $\delta^{18}O$ and $\delta^2H$ ($P < 0.001$) and no significant effect on the isotopic composition of the deep soil water, typical of shallow infiltration-evaporation cycles (Barnes and Allison 1988). In the top soil, soil water content was negatively correlated with $\delta^2H$ ($P < 0.05$), but not with $\delta^{18}O$. This is because changes in the isotopic composition of soil water with rain addition or evaporative losses are several times larger for $\delta^2H$ than for $\delta^{18}O$ (depending on the slope of the LMWL or the SWL). No similar correlation was observed between soil water content and $\delta^2H$ in the deep soil probably because the range of variations

was smaller (Fig. 1 and Fig. 2). Finally, rock moisture was significantly more enriched in both isotopes than top and deep soil, but fell along the LMWL (Fig. 2). The isotopic signal of rock moisture did not differ between plots over time, nor did it correlate with weather conditions or with the isotopic signal of top or deep soil water.

**3.3 Stable isotopes of xylem water**

The isotopic composition of xylem water always fell underneath the LMWL in the dual-isotope space (Fig. 2). Xylem water from the first campaign in late April (i.e. just before or during budburst), was exceptionally enriched (Fig. 3) and displayed a very clear evaporation line in the dual-isotope space (green "outliers" on the right side of Fig. 2). This was indicative of stem evaporative enrichment over winter, as observed in other species (Bowling et al., 2017; Martin-Gomez et al., 2017). Excluding this first campaign, xylem water samples of both *F. sylvatica* and *Q. robur* overall had a more depleted $\delta^2H$ than top and deep

soil water ($P < 0.001$), as illustrated by Fig. 3. Consequently, a large number of the xylem samples did not match any of the considered sources in the dual-isotope space (Fig. 2).

The diameter at breast height of trees (DBH) had a negative effect on both isotopes of xylem water samples ($P < 0.001$). Consequently, dominant trees of *F. sylvatica* had a more depleted xylem water than non-dominant trees ($P < 0.01$ for both isotopes). *F. sylvatica* presented more enriched values of $\delta^{18}O$ ($P < 0.05$) and tended to have more enriched values in

$\delta^2H$ ($P < 0.1$), compared to *Q. robur*. To our surprise, no significant differences were found in xylem water isotopes between the three studied plots. The four trees in which xylem water was extracted from coarse roots (rather than from twigs) showed a significantly more depleted $\delta^2H$ over the whole season ($P < 0.001$), but no significant difference in $\delta^{18}O$, compared to all the other trees (Fig. 4). Overall, xylem water $\delta^2H$ showed different patterns than $\delta^{18}O$.





The isotopic offset between xylem and soil samples was assessed by calculating the SW-excess. On average, xylem
water samples had a SW-excess of -8.40 ± 5.37‰. There were no significant differences in xylem SW-excess between species,
and its seasonal variations were small (Fig. 5). Although canopy position and DBH had no effect on the SW-excess, the type
of sampling had a strong influence, especially for $\delta^2H$, as the SW-excess was significantly more negative in trees whose xylem
water had been sampled from coarse roots as opposed to twigs (Fig. 4). The GLMM fit used to understand the factors driving
the SW-excess across time and space could only explain 16% of the variance (Table S2). Variables such as rainfall, VPD, top
soil $\delta^{18}O$ and $\delta^2H$ or DBH were excluded from the model selected based on the Akaike information criterion (AIC). Top soil
water content had a positive effect on SW-excess, and was the variable with the strongest explanatory power ($P < 0.01$). The
SW-excess was also significantly affected by deep soil $\delta^{18}O$ ($P < 0.05$, negative effect) and rock moisture $\delta^{18}O$ ($P < 0.01$,
positive effect).

## 3.4. Isotopic mixing models

The first set of isotopic mixing models were run only for the dominant trees of *F. sylvatica* and *Q. robur* using both $\delta^{18}O$ and
$\delta^2H$ data. On average, these mixing models indicated that *F. sylvatica* trees used a mix of top and deep soil water, with a
marginal contribution of stream water (Fig. 6). The same mixing models also indicated that *Q. robur* relied mostly on soil
water as well, but had significantly higher contributions from stream ($P < 0.01$) and deep soil water ($P < 0.01$), and
consequently lower contributions from top soil water ($P < 0.001$), compared to *F. sylvatica*. Nonetheless, both species followed
similar temporal patterns (Fig. 6). The dominant and non-dominant *F. sylvatica* trees had similar source contributions, except
for a surprisingly higher relative uptake from stream water of non-dominant trees (Fig. S2). Differences between plots were
not significant (not shown).

In a second step, we focused on the sensitivity of the isotopic mixing models to the observed $\delta^2H$ offset and the dual-
versus single-isotope approach. For this, we only used the isotopic data for dominant *F. sylvatica* trees (N = 15). Correcting
values of xylem $\delta^2H$ for their SW-excess significantly affected the estimated source contributions of *F. sylvatica* (Fig. 7). The
dual isotope model with corrected $\delta^2H$ values estimated a higher contribution of stream water late in the season ($P < 0.001$)
and deep soil water in the summer compared to the dual isotope model with the original $\delta^2H$ values. Only the contribution of
top soil water was marginally affected by this SW-excess correction. The single-isotope approach using only $\delta^{18}O$ estimated
higher contributions of stream water ($P < 0.001$) and lower contributions of deep soil water ($P < 0.001$) than the dual-isotope
approach with uncorrected $\delta^2H$ (Fig. 7). On the other hand, a single-isotope approach using only $\delta^2H$ estimated lower
contributions of top soil water ($P < 0.001$), and higher contributions of stream water ($P < 0.001$) than the dual-isotope approach
with uncorrected $\delta^2H$ (Fig. 7).

The discrepancy in the estimation of source contribution to xylem water of isotope mixing models with different input
data also translated into a contrasting relationship with environmental data (rainfall, VPD and soil moisture). These



relationships are reported in Table S3, separated by source and input data. Overall, the models using a dual-isotope approach but with corrected $\delta^2H$ values, or only $\delta^{18}O$ showed the strongest and most plausible correlations with environmental variables over the growing season. Although the contribution of stream water to xylem water estimated from $\delta^2H$ only led to the best correlations with rainfall amounts and VPD, however the sign of these correlations were opposite in sign to what could be expected. That said, the use of only one isotope was not sufficient to disentangle the contribution of various water sources for some campaigns where the isotopic compositions of the different water sources were too similar (Fig. 8). In these cases, the Bayesian mixing models predicted equal contributions for each of the three water sources considered (e.g. on 4/7 for $\delta^{18}O$ only, Fig. 7).

## 4. Discussion

### 4.1 Potential causes of the $\delta^2H$ offset between xylem and source water

This study analysed an extensive isotopic dataset of xylem water and variability in its potential sources over a growing season. In addition, pitfalls in the use of stable isotopes to identify and quantify plant water sources were explored and reported. Our results support those from recent studies showing a consistent and relatively stronger depletion in xylem water isotopes than expected based on the measured composition of potential water sources (*e.g.* Bowling *et al.*, 2017; Evaristo *et al.*, 2017; Geris *et al.*, 2017). The diversity of methodologies used for the extraction of waters and their isotopic determination in all these studies, including ours, rules out potential analytical bias. Furthermore, isotopic offsets measured between xylem and source water were consistent over time and space (Fig. 2 and Fig. 5). Other field datasets have shown similar isotopic offsets in semi-arid (Dawson & Ehleringer 1991; Bowling et al. 2017) and saline (Lin & Stenrberg 1993) environments, but here we show that it also occurs in temperate deciduous trees growing in a mild oceanic climate. Furthermore, potted plants (Ellsworth & Stenberg 2007; Vargas *et al.*, 2017) and plants in botanical gardens (Evaristo et al., 2017), have also reported isotopic offsets between soil and xylem water and discussed these to some extent. In addition, a number of other studies from tropical (De Deurwaerder et al., 2018), semi-arid (Wang et al., 2017), temperate (Bertrand et al., 2014; Brooks et al., 2010) and northern ecosystems (Geris et al., 2015, 2017) have also reported similar offsets to those observed in our study however, these results and their repercussions for partitioning were not fully explored. Our results show that $\delta^2H$ offsets between xylem and source water complicate the identification of plant water sources and the source contributions estimated by Bayesian isotopic mixing-models (Fig. 7), a finding in contrast with recent studies (Evaristo et al., 2017). The mismatch between xylem and source water isotopes may be caused by three non-exclusive processes: (1) a water isotope separation between bound and mobile water (Brooks et al. 2010), (2) a water isotope fractionation occurring at the soil-root interface (Ellsworth & Stenberg 2007; Vargas et al. 2017) or (3) a water isotope compartmentalization between xylem water and other stem water pools (Zhao et al. 2016). Indeed, surface-water interactions operating at the pore level (Oerter & Bowen, 2017) and varying as a function of particle size (Gaj et al., 2017) or cation content (Oerter et al., 2014) may create isotopic heterogeneity within the soil matrix. The soil



in our study site is sandy, thus the effect of interactions with clay-absorbed cations are likely to be small (Fig. S2). However,
surface-water interactions on quartz silica or carbon-rich materials have also been shown to affect the water isotope
composition of adsorbed water (Richard et al. 2007; Lin & Horita 2016, 2018; Chen et al. 2016). Assuming that cryogenically-extracted soil water represents bulk soil water, a disparity between adsorbed soil water and more mobile water accessible to
the plant would create a mismatch between plant and bulk soil water. However, because adsorbed water is generally more
depleted than bulk soil water (Lin et al., 2018; Lin and Horita, 2016), the more mobile water taken up by the plants should be
more enriched than bulk soil water, i.e., the opposite of what is found in this study.

The water content of rocks was quite high (12%), so differences between highly enriched rock-absorbed and residual
soil water could have played a role in depleting the water pool accessed by roots. If deeper soil layers with a large rock volume
contribute to root water uptake, it is likely that the isotopic mass balance would result in a depletion of plant-accessed water.
However, belowground physicochemical processes affecting xylem water isotopes may be expected to vary over time and
space especially if isotope effects are related to either the soil type, water content and/or the depth of water uptake. In contrast,
we measured a rather consistent offset, driven mainly by $\delta^2H$ and poorly explained by environmental variables. Similar results
in the literature (although scarcely discussed) can be found at sites with contrasting plant and soil types, soil moisture regimes
and lithology (Brooks et al., 2010; De Deurwaerder et al., 2018; Geris et al., 2017; Vargas et al., 2017; Zhao et al., 2016). Thus
although, the evidence for $\delta^2H$ fractionation between the bulk and plant-accessed soil water pool is growing, the number of
studies probing the mechanisms behind this pattern are still limited. The exact mechanism remains elusive even in the present
study on *F. sylvatica* and *Q. robur* given the weak spatiotemporal variability in SW-excess$_x$ (Fig. 5). Furthermore, if isotopic
fractionation was occurring within the unsaturated zone, we propose that the xylem-source $\delta^{18}O$ offsets should be similar to
those of $\delta^2H$, however this was not the case for our study. Instead we found a significant offset for $\delta^2H$ between xylem and
water sources for all our trees, with an even stronger depletion when the xylem water was collected from coarse roots (Fig. 4).
Branch evaporation could underlie this difference, such as in the case of branch evaporative enrichment, however, again we
should observe a similar offset for both $\delta^2H$ and $\delta^{18}O$ (Martín-Gómez et al., 2017). Previous studies have shown that water in
coarse or tap roots can exhibit significant depletion in $\delta^2H$ relative to source water pools fore examples in *Populus euphratica*
(Zhao et al., 2016) and *Prosopis velutina* (Ellsworth and Williams, 2007). Moreover, the $\delta^2H$ offsets reported between soil and
root water were of the same order of magnitude (ca. -20‰ for *P. euphratica* and ca. -7‰ for *P. velutina*) as observed for *F.
sylvatica* and *Q. robur* in this study (Fig. 4 and Fig. 5). Interestingly, Zhao *et al.*, (2016) analysed xylem water and what they
called tissue water (*i.e.* bulk plant water) separately with the former not showing any isotopic depletion compared to source
water. In this context, Ellsworth & Williams, (2007) attributed the $\delta^2H$ depletion in halophytic plants to their strong dependence
on the symplastic pathway for water movement into the root. Sympastic transport of water is not only important during root
water uptake, but also plays a relevant role in the wood capacitance of *Euclayptus saligna*, notably during periods of increasing
tension in the xylem (Pfautsch et al., 2015). This pathway uses ray parenchyma cells and is used to maintain a two-way
exchange of water between the phloem and the xylem. It is thus plausible that in roots, containing higher fractions of





parenchyma cells than stems (Morris et al., 2016), a relatively higher volume of water moving through the symplast could cause a strong depletion of bulk wood water, the water sampled during cryogenic extraction. For the species in this study, ray and axial parenchyma can account for around 31% of total xylem tissue in both *F. sylvatica* and *Q. robur* (Morris et al., 2016). Thus future studies are now required to explore this potential mechanism that could provide valuable insights on the depletion

in $\delta^2H$ of water transported through the symplast. In addition, this mechanism may be quantitatively relevant for interpreting the isotopic composition of bulk xylem water, as the use of water stored in the stem can account for up to 16% in *F. sylvatica* (Köcher et al., 2013), and contribute even more in some subtropical tree species (Oliva Carrasco et al., 2015).

### 4.2. Estimation of source contribution to xylem water

The Bayesian isotope mixing-model demonstrated that both *F. sylvatica* and *Q. robur* use a mixture of both top and deep soil water throughout the growing season, with marginal contributions of streamwater (or groundwater) (Fig. 6). The use of tree water sources was relatively plastic, as shown by rapid shifts towards the deep soil in late summer when soil was progressively drying. Overall *Q. robur* used more deep soil water than *F. sylvatica*, probably because the root distribution in the former species tends to be deeper (Rosengren et al., 2005), whereas the latter species usually has a denser fine root network in top soil

layers (Leuschner et al., 2001). Typically, *Q. robur* is distributed throughout the region over acidic sandy soils containing a hardpan that restricts access to the groundwater. In contrast the riparian forest in the present study was located on a slightly more basic soil (Table S1) with a rocky layer of limestone that provided a water connection to the groundwater. This may be a reason why *F. sylvatica* is consistently restricted to a few particular sites in its southern range (Lafontaine et al., 2014; Timbal and Ducousso, 2010). In addition, our results and those from other studies indicate that *F. sylvatica* typically have

shallower rooting depths and use relatively shallower soil water sources throughout the year. Collectively, these results support the hypothesis that the persistence of *F. sylvatica* in climate refugia could be linked to edaphic conditions that may be enhanced by the presence of rocky layers of limestone permeable to water and providing a potential geologic reservoir of water for tree fine roots and fungal hyphae to tap into.

Finally studies applying isotope mixing-models such as *MixSIAR* to study plant water sources usually set the

discrimination factor between source and xylem water to zero (Barbeta et al., 2015; Evaristo et al., 2017; Palacio et al., 2014; Rothfuss and Javaux, 2016). However, as we report here, $\delta^2H$ fractionation may occur either within soil water pools or within plant tissues, impacting the measured isotopic composition of cryogenically-extracted xylem water. Our approach to evaluate the potential effects of isotopic fractionation between the soil and the plant consisted of comparing models with different input data. By correcting xylem $\delta^2H$ based on the $SWexcess_x$, we found that the contribution of stream water had been

underestimated compared with the classic approach (Fig. 7). Contrary to recent studies that reported a low sensitivity of Bayesian isotope mixing-models to $\delta^2H$ fractionation (Evaristo et al., 2017), we found that the plant-water source estimations also varied between models using either $\delta^{18}O$ only, $\delta^2H$ only or models using both isotopes. In addition, this disparity caused by $\delta^2H$ fractionation cannot be solved by using only the apparently non-fractionating $\delta^{18}O$ as having only one dimension is not



enough to distinguish between water sources in many cases (Fig. 8). However, Bayesian mixing-models were shown to perform

best in a recent comparison of approaches to quantify root water uptake (Rothfuss and Javaux, 2016). Although it is worth noting that these comparisons are based on the assumption that no fractionation occurs. Therefore, the application of these models may not be suitable in studies where $\delta^2H$ fractionation is suspected. Interestingly, we found that once the model assumptions were corrected the $\delta^2H$ data then gave strong correlations with environmental data and allowed for a more parsimonious interpretation (*e.g.* 5-day rainfall amount positively correlated with top soil water uptake) (Table S3). Based on

these correlations, correcting xylem water isotopes using SW-excess$_x$ appeared to improve the power of the dual-isotope approach. Specifically, we corrected for potential fractionation affecting $\delta^2H$ by re-aligning xylem water isotopes with the soil water line, however xylem water might not necessarily plot right on this line if $\delta^2H$ fractionation does not occur.

## 5. Conclusion

In light of our results and other recent studies either conducted under controlled conditions (Oerter et al., 2014; Vargas et al., 2017) or in the field (Evaristo et al., 2017; Oerter and Bowen, 2017; Oshun et al., 2015; Zhao et al., 2016) evidence for fractionation processes occurring within the unsaturated zone, the soil-root interface or within plant woody tissues is growing. These processes may complicate or prevent the identification of plant-water sources, especially when they remain unnoticed. Importantly, $\delta^2H$ fractionation during or after root water uptake seems to extend beyond plants growing in salty and dry

environments (Ellsworth and Williams, 2007). This should now motivate researchers to develop hypothesis-driven studies focused on two main lines. Firstly, to couple the study of physicochemical fractionation processes in the unsaturated zone with their repercussions on plant absorbed water, covering a range of soil properties and water content (as illustrated by Vargas *et al.*, (2017)). Secondly, obtain a better understanding of the isotopic dynamics of potentially different water pools within plant tissues. In particular, those tissues that actively transport storage water back and forth between parenchyma cells and

conductive tissues (Pfautsch et al., 2015). Interestingly, if there are two isotopically distinct water pools, this could be even explored to quantify the contribution of wood water storage use in conferring wood capacitance. On the other hand, the appropriateness of Bayesian isotope mixing-models to partition plant water sources depends on whether $\delta^2H$ fractionation is taking place or not. However, establishing whether isotopic fraction between xylem and its sources is occurring can be easily assessed analysing the dual-isotope space. It can also be quantitatively assessed by calculating the SW-excess$_x$, that will

produce negative values when xylem is more depleted than the soil and positive values when xylem is more enriched than the soil. In some cases this can be interpreted as having a plant-water source missing (Bowling et al., 2017), however we show here that this is not necessarily the case given the fractionation processes described above. In the future, a more detailed mechanistic understanding of $\delta^2H$ fractionation could help correct the $\delta^2H$ of xylem water more accurately and provide more parsimonious water source estimations in cases such as the study presented here. Furthermore, *MixSIAR* and other Bayesian

isotope mixing-models can be adapted easily to account for $\delta^2H$ discrimination between sources and xylem water if this is





quantitatively assessed beforehand, as is currently the case in other applications such as in food web studies (Phillips et al., 2014).

Although the present dataset does not allow us to assess definitively which are the ecohydrological mechanisms that have assisted the persistence of *F. sylvatica* in this riparian forest for at least the last 40 thousand years, we can rule out a prevalence for water uptake from the stream, as found in other riparian tree species (Bowling et al., 2017; Dawson and Ehleringer, 1991). This is also in agreement with the fact that the regionally common *Q. robur* seems to have a relatively higher use of deeper soil water than *F. sylvatica*, with typically shallower root systems. However, this river valley is characterised by a relatively shallow layer of weathered limestone that may confer different and advantageous soil characteristics compared to regions, where *F. sylvatica* is absent. Indeed, the C horizon is located within the rooting depth of

both studied species (at a depth 50-120 cm) and has a higher clay fraction as well as more fine sand and less coarse sand than the horizons A and B, that are more representative of the region's soil profiles. Therefore, a higher water holding capacity of soils around the Ciron river could be responsible for the long-term persistence of *F. sylvatica* in this valley. Stable isotope techniques are just one tool that can help explore mechanisms sustaining climate refugia, and more particularly, hydrological refugia (McLaughlin et al., 2017). By comprehensively studying belowground water dynamics with multiple tracers, isotope-

enabled land-surface models that have the ability to resolve and simulate the impacts of different soil and plant root characteristics on the exchange of water, carbon and energy between the soil-plant-atmosphere continuum can help predict the fate and distribution of climate and hydrological refugia in the future.

*Data availability*

The data collected in this study is available upon request to the authors.

*Author contributions*

A.B., J.O. and L.W. designed the study. A.B., L.C., B.F., S.J., T.E.G. and S.W. conducted the field work. S.J. and S.W developed and tested the cryogenic water extraction vacuum line, and produced the protocols and code to process water isotope

data. B.F. and L.C. ran all the samples through the water extraction line. A.B., L.C., B.F., and S.J., conducted the water isotope analyses. A.B. and T.E.G analysed the data, and all the authors contributed to the discussion of the results obtained. A.B. wrote a first draft of the manuscript, that was edited in several rounds of revision by all authors.

*Competing interests*

The authors declare that they have not conflict of interest.

*Acknowledgements*

This study has been carried out on the HydroBeech, ClimBeech & MicroMic, projects with financial support from the French National Research Agency (ANR) in the frame of the Investments for the future Programme, within the Cluster of Excellence





COTE (ANR-10-LABX-45). A.B acknowledges a IdEx Bordeaux Postdoctoral fellowship from the Université de Bordeaux (contract no. 22001162). This project has also received funding from the European Research Council (ERC) under the European Union's Seventh Framework Program (FP7/2007-2013) (grant agreement no. 338264) awarded to L.W., a Marie Skłodowska-Curie Intra-European fellowship (Grant Agreement No. 653223) awarded to T.E.G., the French Agence National de la Recherche (ANR) (grant agreement no. ANR-13-BS06-0005-01) awarded to J.O. and the Aquitaine Region project

Athene awarded to L.W, (2016-1R20301-00007218).

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




**Figure 1: Environmental conditions during the growing season of 2017 in the riparian forest in the river Ciron. From the top panel**
**to the bottom; daily vapor pressure deficit (VPD), streamflow of 2017 plotted along the 2010-2017 average, daily precipitation and**
**gravimetric water content for the top soil (0-10 cm) and the deep soil (ranging from 50 to 120 cm).**








**Figure 2: Dual isotope (δ²H and δ¹⁸O) plot of xylem water and its potential sources (soil al two different depths, stream and groundwater, rock moisture and rain water) for every sampling campaign conducted in 2017. The blue corresponds to the LMWL whereas the dashed black line correspond to the GMWL.**






**Figure 3: Boxplots of the isotopic composition of top soil water (0-10 cm), deep soil water (50-120) and xylem water of the two studied species (*F. sylvatica* and *Q. robur*) for each single campaign. Data is pooled over the three studied plots. Box size represents the interquartile range, the black line is the median, the whiskers indicate variability outside the upper and lower quartiles, and individual points are outliers.**







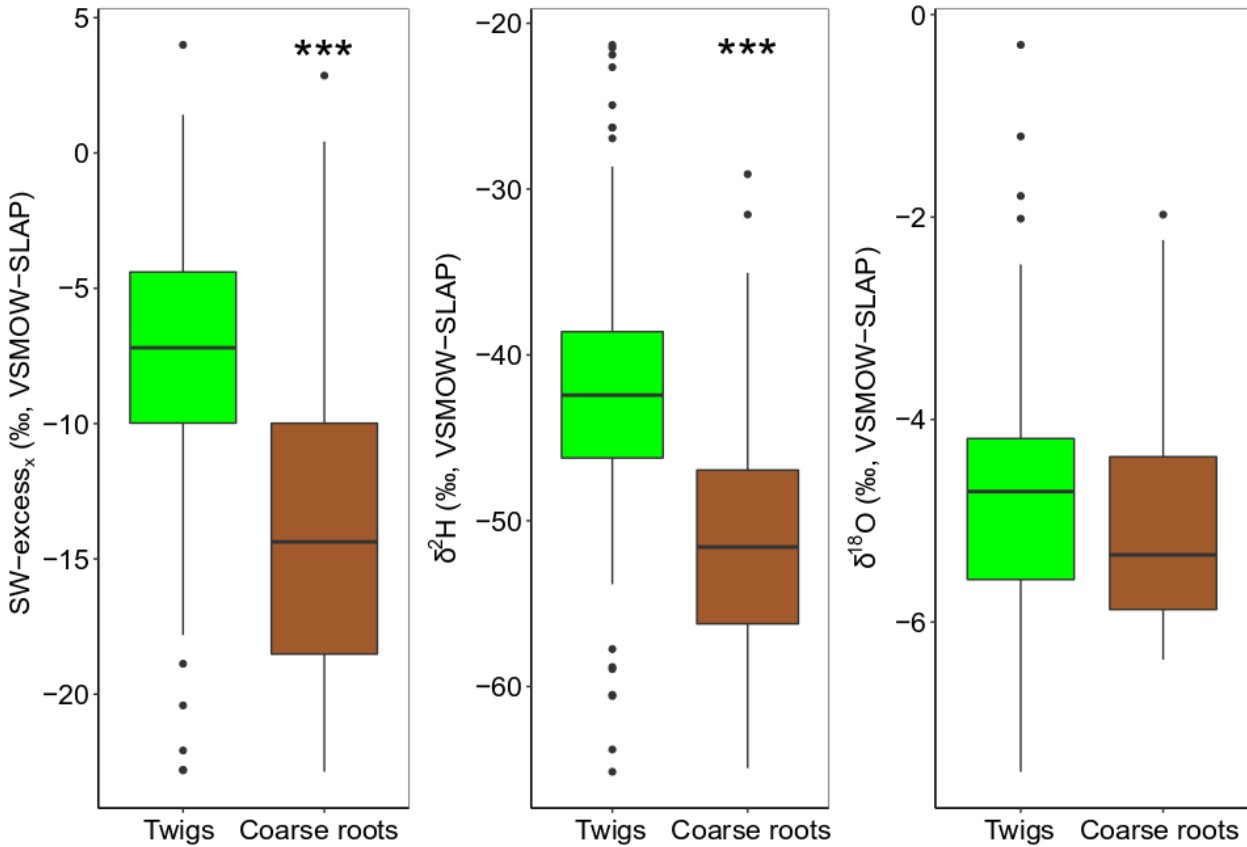

**Figure 4: Boxplot for the differences in SW-excess$_x$, δ$^2$H and δ$^{18}$O of xylem samples of dominant *F. sylvatica* and *Q. robur* depending on the part of the tree sample (twigs or coarse roots). Significant differences (*P*<0.001) are highlighted with asterisks (***). Box size represents the interquartile range, the black line is the median, the whiskers indicate variability outside the upper and lower quartiles, and individual points are outliers. Xylem samples from the first sampling campaign were excluded from the the analysis and the plot because of probable winter branch evaporation.**









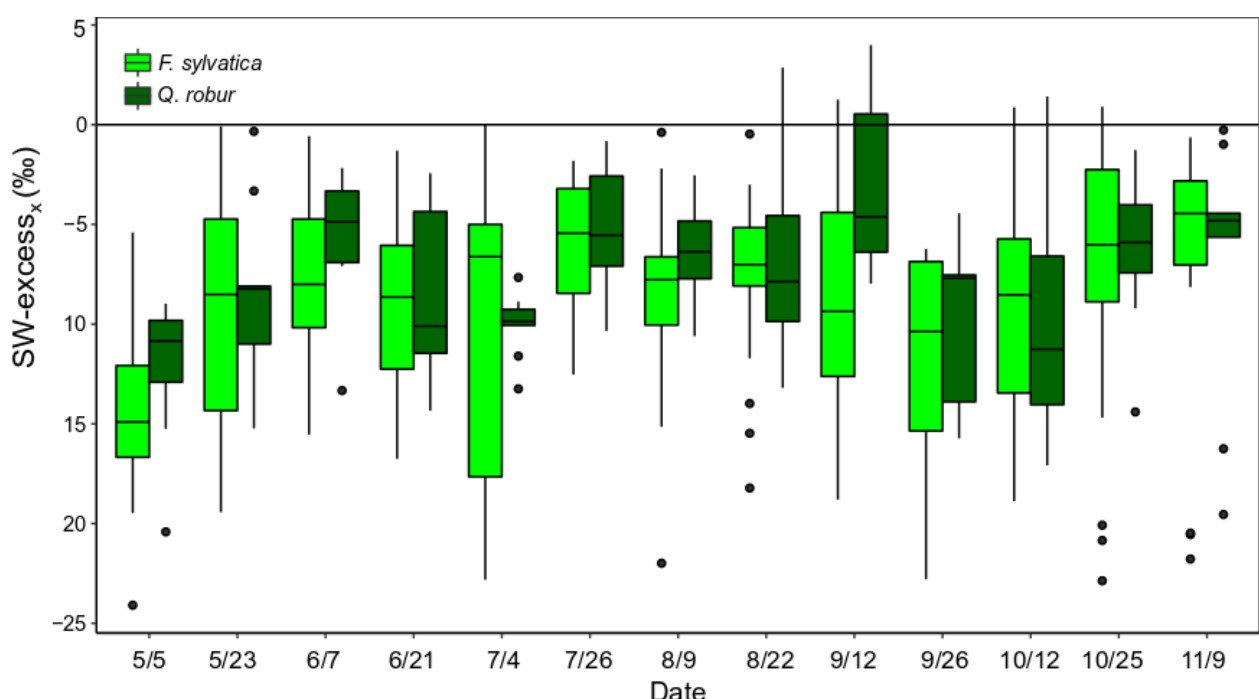

**Figure 5: Boxplot of the temporal variation of in SW-excessₓ of *F. sylvatica* and *Q.* robur xylem samples over the growing season of 2017. Box size represents the interquartile range, the black line is the median, the whiskers indicate variability outside the upper and lower quartiles, and individual points are outliers.**



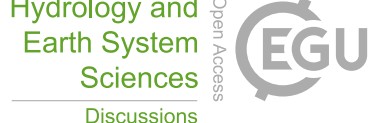

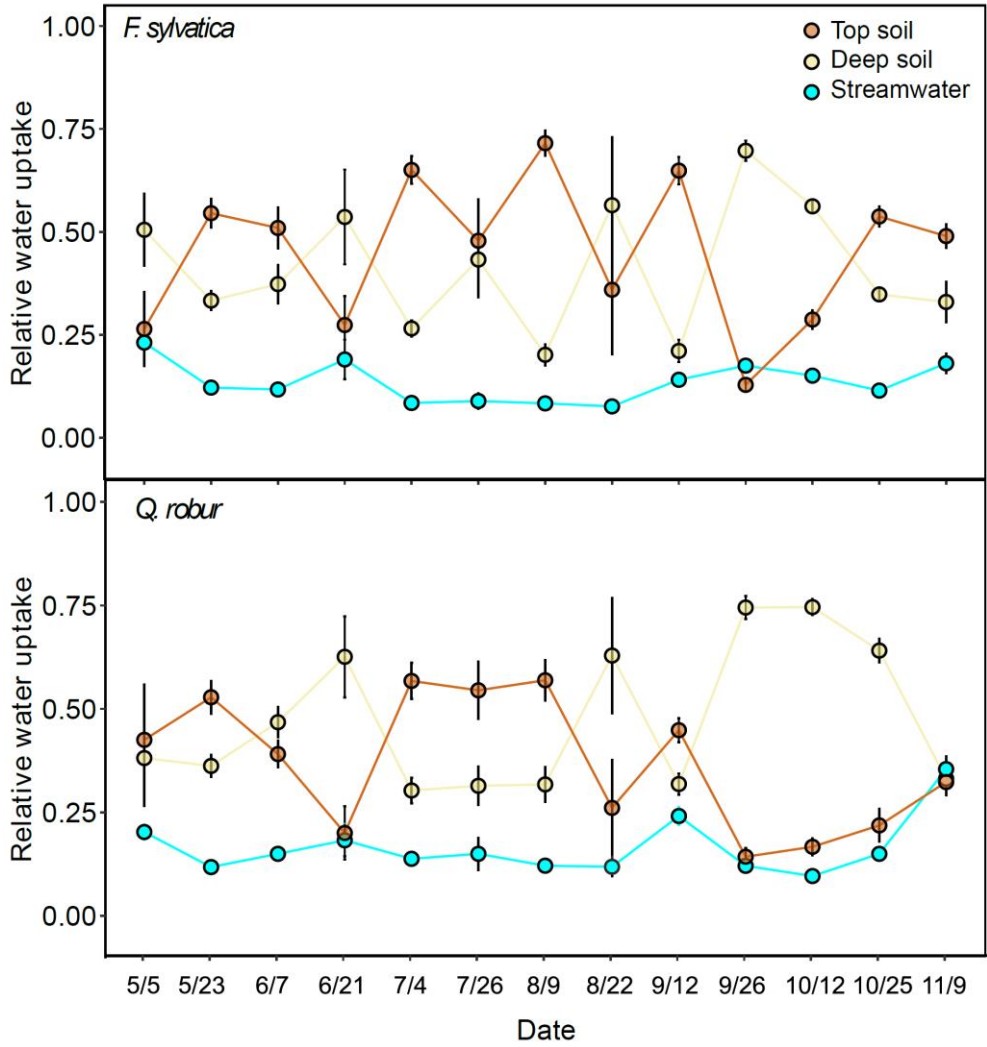


**Figure 6: Relative water uptake from each of the plant-water sources considered: top soil, deep soil and stream water (indistinguishable with groundwater), as estimated with *MixSIAR* for the dominant trees of *F. sylvatica* (top panel) and *Q. robur* (bottom panel). The error bars correspond to the standard deviation (N=15 for *F. sylvatica* and N=9 for *Q. robur*). This proportions were estimated with uncorrected $\delta^2$H and $\delta^{18}$O values for xylem water.**







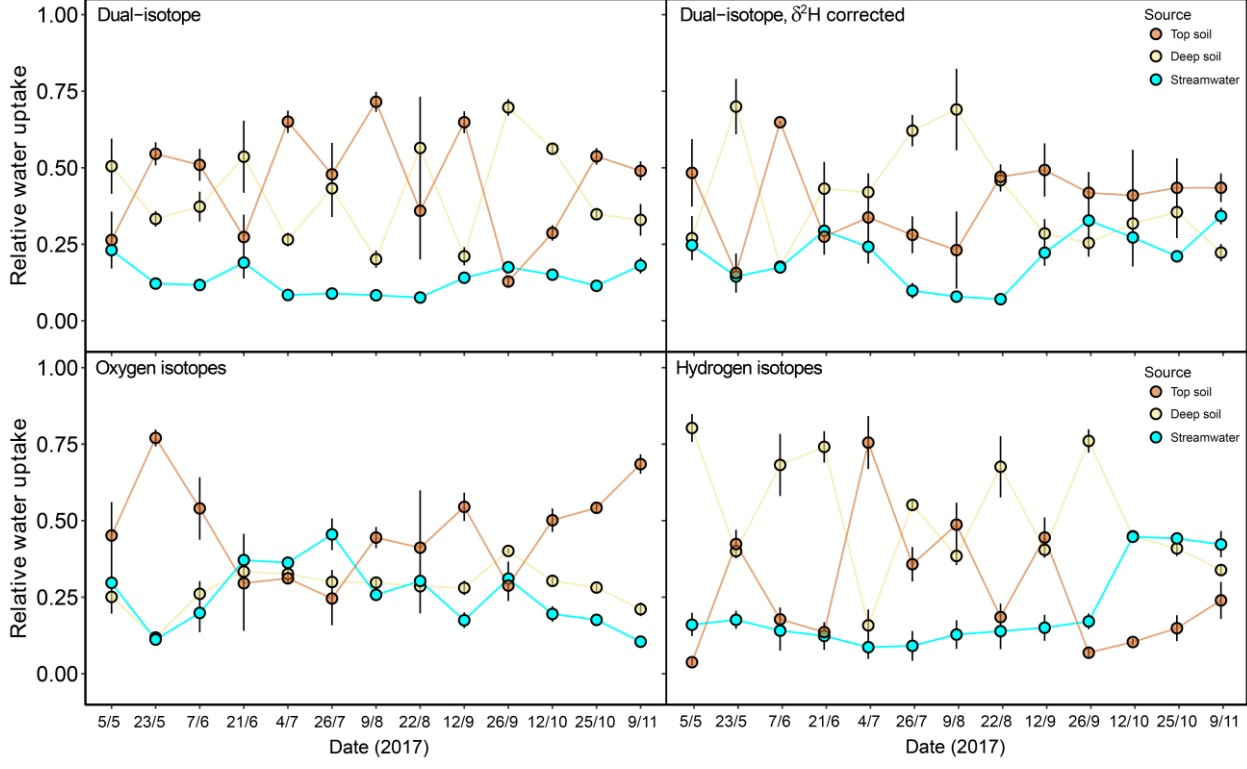

**Figure 7: Relative water uptake from each of the plant-water sources considered: top soil, deep soil and stream water**
**(indistinguishable with groundwater), as estimated with *MixSIAR* for the dominant trees of *F. sylvatica*. The input data is different**
**for each of the four panel, top left; uncorrected $\delta^2$H and $\delta^{18}$O, top right; $\delta^2$H (corrected with the SW-excess$_x$) and $\delta^{18}$O, bottom right;**
**only $\delta^{18}$O and bottom right; only $\delta^2$H. The error bars correspond to the standard deviation (N=3).**







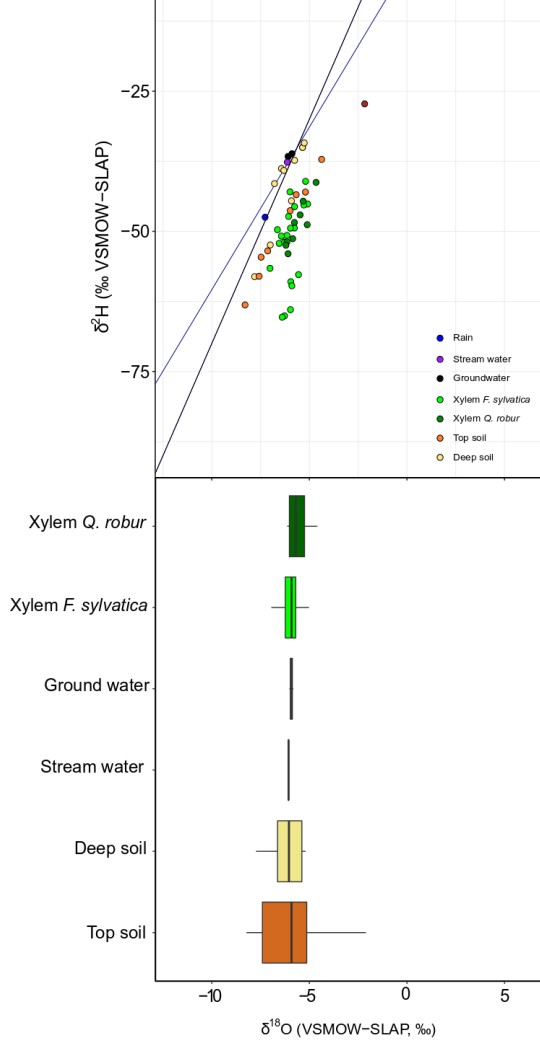

**Figure 8: Differences in the isotopic composition of xylem water and its potential sources using one isotope. The top panel depicts the dual-isotope plot for a single date (July 4[th]), with xylem water and sources. The bottom panel is the boxplot of the $\delta^{18}O$ values for xylem water of *F. sylvatica* and *Q. robur* and each of the potential sources. Box size represents the interquartile range, the black line is the median, the whiskers indicate variability outside the upper and lower quartiles, and individual points are outliers.**