# Peer review of "Hydrogen isotope fractionation affects the identification and quantification of tree water sources in a riparian forest"

_Hydrology and Earth System Sciences, 2018_

## Referee Comment (RC1) · M. Sprenger (Referee) · 31 Aug 2018

**General comments**

Barbeta et al. present a very interesting and extensive stable isotope data set studying the ecohydrology of riparian forests in SW France. Their experimental design of fortnightly sampling over an entire vegetation period is very timely and has pioneer character as most studies addressing similar research questions were limited to few sampling campaigns.

They use well-established methods, both for data gathering (e.g., cryogenic extraction) and data (e.g., MixSIAR) and assess both critically. Their findings regarding a potential

deuterium offset and its consequences for plant water uptake studies is timely and will be of interest to a wide range of scientists working with stable isotopes to investigate soil-plant interactions. Their study therefore fits to the scope of HESS and suggest publication after a minor revision addressing the issues raised below.

I am not sure if the title is correct, when talking about "Hydrogen isotope fractionation", as you cannot prove that it is a (natural) fractionation process, as there could be also methodological issues causing a mismatch between potential sources and plant water stable isotopes (as you discuss). Maybe "Deuterium offset affects..." is more appropriate.

There is a need for a more critical assessment of the sw-excess correction of xylem data. You refer to Bowling et al. (2017) in the conclusion (xylem water as a potential result of mixing of enriched top soil water and depleted subsoil water), when discussing alternative explanations for xylem water samples plotting below soil water samples in the dual isotope plot. This should be extended as the presented deuterium offset cannot be explained with the current field experiments. Further, xylem waters can also plot above the soil water samples in the dual isotope due to mixed water uptake of soil water and recent infiltrated isotopically enriched rain water. Correction with the sw-excess would not make sense in this case for example.

**Specific comments**

L 41: Unclear how a climate refugia gets "buffered".

L 54: "the soil depth reached by infiltrating water" reads as if infiltrating water has some kind of boundary beyond which it cannot further percolate. I suggest to refer to mixing processes of water in the subsurface instead.

L 51: To clarify that you refer to sampling and not root accessibility, I suggest to say "... and can be sampled" (or similar).

L 94: However, Oerter et al. (2014) that you quote in L 89 actually showed this fractionation for clay minerals.

L 96: This is unclear. If there is only pore water in small pores, then there cannot be

any interaction with mobile water, as it is not present.

L 145: How is the climate defined?

L 229: How was that tested?

L 252: How did you get volumetric soil moisture? I only see gravimetric moisture described in the methods.

L 253: How, where, and how often was the rock water content measured?

L 263 and following: It is not clear which statistical tests were applied, as they are not specifically mentioned in the methods section.

Fig. 2: Which rain data is shown? Rain that fell on the given days? Rain that fell between the 14 days previous sampling? If so, are they weighted averages?

You mention rock moisture in the caption, but I do not find it in the legend.

L 266: Not clear if the SWL was calculated for only topsoil or all soil samples.

L 269: Negative effect means more depleted in heavy isotopes with higher antecedent rainfall? Does the isotopic compositions of the antecedent rainfall correlate with the top soil isotopic compositions?

L 272: This explanation is not clear to me.

L 276: I do not see the rock moisture isotope data in Fig. 2.

L 282: Please reword, as "evaporation line" is evaporation line is defined as the linear change in the isotopic composition in dual isotope space of a single water source through evaporative concentration of the remaining liquid. However, these xylem samples were taken all at the same time. I do not question that they underwent evaporation and kinetic fractionation, but "evaporation line" is the wrong word.

L 286: Needs rewording, as "had a negative effect" implies that actually DBH influences the xylem water isotope compositions. I suggest saying something like "more depleted with higher DBH".

L 291: This questions that grouping xylem and root samples in Fig. 2, Fig. 3, and Fig. 5. You should then present each data set separately there already. Or do you not present any extracted root water there? Not clear to me.

L 306: What does "only for the dominant trees" mean? Why not all sampled trees?

Table S3: I encourage the authors to add this Table S3 to the main manuscript, as this presents a major finding of the study. Why did you decide to take 5-day averages and amounts?

L 354: The "water isotope fractionation occurring at the soil-root interface" would not cause the deuterium offset, as Vargas et al. (2017) describe it as a Raleigh type fractionation.

L 360: You mention a carbonate rich C Horizon in the methods. Could processes described by Meißner et al. (2014) be relevant here?

L 362: I would disagree with this generalization of more tightly bound water being generally more depleted, as it seems to be depending on the time of sampling: During summer, mobile water will be more enriched in heavy isotopes as the rainfall is relatively enriched, as the more tightly bound water, which is likely to have a isotopic composition that stems from winter (more depleted) rainfall/snowmelt. I discussed this issue in Sprenger et al. (2018). However, as most field sampling takes place during summer, various studies showed more isotopically depleted waters in more tightly bound water than mobile water (Geris et al., 2015; Oerter and Bowen, 2017).

L 375: Why "weak spatiotemporal variability"? The variability is partly very high (see F. sylvatica on July 4). It's the pattern that is weak, which the current study cannot really well explain.

L 385: In the light of Zhao et al. (2016) and your discussion in L 390 and following, do you then think that cryogenic extraction is then the best way to sample xylem water? Consider discussing potential impacts of not limiting the sampling to the actual transpiration water when using cryogenic extraction. However, why is sw-excess then so variable?

L 429: As you pick up the Table S3 again as one of the main findings, I highly suggest including it in the main manuscript and not having it in the supplementary material.

L 432: This is a very important point you raise. How did you deal with xylem data of positive sw-excess? This could be a result of a mixed source (soil water plus recently infiltrated rain water). I can also imagine a situation, where an enriched soil water

source in the top soil with a depleted source (e.g., snowmelt) in deep soil results in a xylem water isotope composition plotting below the sw-line, but does not result from deuterium fractionation. Consider discussing the limitations of your sw-excess correction.

**Technical corrections**

L 18: No delta sign (applies to the entire abstract)

L 18: bi-weekly could mean twice a week or every two weeks (fortnightly); applies also to L 154 and L 259.

L 31: This is not a question. You either need to remove the question mark or rephrase.

L 39: ecohydrological

L 41: understanding

L 46: Somewhere here you need to introduce "stable isotopes of water (2H and 18O)", as you currently miss this, but use 2H and 18O it in L 67 without having it introduced.

L 68: No hyphens

L 69: . . .rainless summer fills small soil pores first and does not contribute to river flow nor to mixing. . .

L 74: TWW hypothesis holds, . . .

L 78: . . .stream water for riverside trees.

L 96: . . .when only water in small pores is accessible. . .

L 97: . . . between sampled soil pore and . . . (important to mention "sampled", as this match is solely based on H1 and H2)

L 122: "to the data sets" appears to be more confusing than clarifying.

L 146: 813 mm/year (no digit needed and the unit is mm/year)

L 191: composition (given in delta notation . . .)

L 214: I suggest to not use sigma, but a different letter, since it could be misinterpreted as standard deviation.

L 224: were run

Fig. 1: It seems that there is something wrong with the x-axis.

The VPD data is shown as a continuous measurement, but the x-axis is not really a

time axis, as the x-axis ticks are not equally distributed (e.g., every 14 days). Not sure if the date format is correct according to the Copernicus guidelines, and it's actually different format to Fig. 2.

The streamflow data does not look like actual data per second, but seems to be an average value for the individual days. If so, this should be mentioned. Add mm/day for precipitation.

Thus, GWC should have a different x-axis as the other three subplots.

L 260: I suggest defining delta notation in the methods and then you do not need to refer to VSMOW-SLAP scale here.

Fig. 2: I encourage the authors to consider using in addition to different colors also different markers for e.g., soils, vegetation, precipitation,... just to make it easier to see the differences in the dual isotope plots. Please also consider increasing the marker size. I suggest defining delta notation in the methods and then you do not need to refer to VSMOW-SLAP scale here. Typo: "...soil at two..." and "The blue line corresponds...black line corresponds to..."

Fig. 3: Consider adding "cm" to 50-120. I suggest defining delta notation in the methods and then you do not need to refer to VSMOW-SLAP scale here. Not sure if the date format is correct according to the Copernicus guidelines, and it's actually different format to Fig. 2.

Fig. 4: Please use different colors, as this is misleading when similar colors are used in Fig. 3 even though there is different data shown (soil vs roots). I suggest defining delta notation in the methods and then you do not need to refer to VSMOW-SLAP scale here.

Fig. 6: Why hyphenated plant-water?

References

Bowling, D. R., Schulze, E. S., and Hall, S. J.: Revisiting streamside trees that do not use stream water: Can the two water worlds hypothesis and snowpack isotopic effects explain a missing water source?, Ecohydrol., 10, e1771, doi:10.1002/eco.1771, 2017.

Geris, J., Tetzlaff, D., McDonnell, J., Anderson, J., Paton, G., and Soulsby, C.: Ecohydrological separation in wet, low energy northern environments?: A preliminary assessment using different soil water extraction techniques, Hydrol. Process., 29, 5139–5152, doi:10.1002/hyp.10603, 2015.

Meißner, M., Köhler, M., Schwendenmann, L., Hölscher, D., and Dyckmans, J.: Soil water uptake by trees using water stable isotopes ($\delta$2H and $\delta$18O)$-$a method test regarding soil moisture, texture and carbonate, Plant Soil, 376, 327–335, doi:10.1007/s11104-013-1970-z, 2014.

Oerter, E. and Bowen, G.: In situ monitoring of H and O stable isotopes in soil water reveals ecohydrologic dynamics in managed soil systems, Ecohydrol., 10, e1841, doi:10.1002/eco.1841, 2017.

Oerter, E., Finstad, K., Schaefer, J., Goldsmith, G. R., Dawson, T., and Amundson, R.: Oxygen isotope fractionation effects in soil water via interaction with cations (Mg, Ca, K, Na) adsorbed to phyllosilicate clay minerals, J Hydrol, 515, 1–9, doi:10.1016/j.jhydrol.2014.04.029, 2014.

Sprenger, M., Tetzlaff, D., Buttle, J. M., Laudon, H., Leistert, H., Mitchell, C. P. J., Snelgrove, J., Weiler, M., and Soulsby, C.: Measuring and modelling stable isotopes of mobile and bulk soil water, Vadose Zone J, 17, 170149, doi:10.2136/VZJ2017.08.0149, 2018.

Vargas, A. I., Schaffer, B., Yuhong, L., and Sternberg, L. d. S. L.: Testing plant use of mobile vs immobile soil water sources using stable isotope experiments, New Phytologist, 215, 582–594, doi:10.1111/nph.14616, 2017.

Zhao, L., Wang, L., Cernusak, L. A., Liu, X., Xiao, H., Zhou, M., and Zhang, S.: Significant Difference in Hydrogen Isotope Composition Between Xylem and Tissue Water in Populus Euphratica, Plant Cell Environ, 39, 1848–1857, doi:10.1111/pce.12753, 2016.

---

## Referee Comment (RC2) · Xin Song (Referee) · 7 Sep 2018

In this study, Barbeta et al. applied stable isotope techniques to investigate potential water sources of two broad-leaved tree species in a temperate, riparian forest. For this purpose, they made collection of a season-long dataset of d18O and dD compositions from tree xylem water, soil water at different depths, and other potential water sources. They show for both of the tree species that different water sources can be appropriately identified with oxygen isotope data. However, the same conclusion cannot be drawn from hydrogen isotops, as dD of xylem water on many occasions apparently fell out of the range as encompassed by the potential sources. The authors made a detailed

discussion of several possible causes of the observed xylem-soil water dD mismatch, and concluded that isotopic fractionation in the unsaturated zone and/or within plant tissues could well be the driving mechanism.

This is a well-designed field study that addresses an important topic in stable isotope ecohydrology. The manuscript is well written, and data analysis was carried out in a solid manner. The finding of dD fractionation in two commonly occurred temperate tree species is a timely reminder that this phenomenon may not be an exception as previously thought, only restricted to a narrow range of species such as mangroves or species from arid regions, but more likely one that is common to a wider range of species. This finding will have important implications for stable isotope ecohydrological and ecophysiological studdies.

1. Line 232: There is a lack of explanation of the rationale behind the modeling exercise of using SW-excess corrected dD. As far as I understand, given the possibility that fractionation occurs at the soil-root interface or within plant tissues, the purpose of applying such type of correction is to obtain real dD values of the water that is available for uptake by plant roots (i.e. correcting dD back to the point before fractionation occurs). If this is the case, then an apriori assumption for doing correction based on a SW-excess line would be that soil water pools are the only sources of water available for tree roots to take up (otherwise an observed "apparent" SW-excess could have been caused by contribution from non-soil sources that do not necessarily follow the SW-line). Was such an assumption met in the present study? How general applicable is this sw-excess based method to other studies? I think it is worthwhile for the authors to discuss further on these important points (this could be done either in M&M or Discussion).

2. Line 272: I understand the authors' argument for a lack of sensitivity of d18O to soil water content here, but still the line "changes in the isotopic composition of soil water with rain addition..." seems somewhat contradictory to what is already stated in Line 269 "rainfall amount...had a negative effect on tope soil water d18O...". Isn't it like

saying that "rain addition has a significant effect on d18O" versus "rain addition may not cause sufficient changes in d18O"?

3. Line 164: How often was stream and groundwater collected? This sentence reads as if they were collected every day? Or on every sampling campaign?

4. Line 276: where is the rock moisture data in Fig. 2?

5. Lines 280-284: I'd like to argue here that if xylem water had become progressively enriched due to stem evaporative enrichment over the past winter, we would expect xylem water to deviate not only from the LMWL line but also from SWL. Yet, from fig. 2 soil and xylem water appear to fall into pretty much the same line.

6. Line 346: change "have also reported isotopic offsets" to "have also been reported to display isotopic offsets"

7. Lines 360-361: Is there also a possibility that cryogenically extracted soil water does not truly represent bulk soil water? See a recent Ecohydrology paper by Orlowski et al. (2016) Critical issues with cryogenic extraction of soil water for stable isotope analysis

8. Line 379: change the second comma to semicolon

9. Line 381: "fore example" should be "for example"

10. Line 387: "sympastic" should be "symplastic"

11. Line 444: this is not a complete sentence

12. Line 445-446: The idea is great, but may not be easy to realize with the current extraction based method that is only capable of extracting bulk water from a plant tissue. Methodological advancement is apparently needed to confer ability of separately analyzing water from different pools (i.e., parenchyma cells versus xylem water) within a given plant sample.

13. Line 448: change "fraction" to "fractionation"

14. Fig. 5: minus signs are missing in several places of the y-axis

regards,

Xin Song
* * *

---

## Referee Comment (RC3) · Y. Rothfuss (Referee) · 10 Sep 2018

In their manuscript submitted to HESS and currently in its discussion phase, Barbeta and colleagues investigate the water sources of F. sylvatica and Quercus robur (L.) on basis of water stable isotopic measurements in sap xylem, stream/groundwater, and soil water in a vertical profile. For this, they opt for the most commonly used method, which is inversing the isotopic data with statistical modeling. They choose MixSIAR, a mixing model embedded in a Bayesian framework. A particularity of this study is that xylem water isotopic samples of both species very often not plot on the soil evaporation line in a ($\delta$18O, $\delta$2H) coordinate system, which is explained by the authors as being due

to a significantly higher hydrogen than oxygen isotopic fractionation during root water uptake.

The manuscript is generally well written (aside sometimes from the isotopic terminology – see my specific comments), easy to follow, and falls into the scientific scope of HESS.

My general concern is that there is a major contradiction between the first part of hypothesis H2 ("it is essential that all potential water sources are identified and accessible") and the fact that the authors deliberately do not sample from the soil between 10 and 70 (or 110, "depending on the depth of the rocky layer") cm depth. In a review article (Rothfuss and Javaux, 2017), a simple synthetic experiment showed how much the non-fulfillment of H2 could lead to biased estimates of the relative contributions to plant root water uptake across a set of potential sources. If the authors can provide proof that the "mid-soil" is isotopically not any different than another source (e.g., deep soil), then it can be pooled together with this other source (e.g., "mid-deep soil"). If this is not the case, the authors' analysis might not be valid. For instance, we cannot say for sure that the observed hydrogen isotopic fractionation is not in fact partly due to the values of isotopic compositions in the missing soil water source.

Cheers,

Youri Rothfuss

Some specific comments follow below:

L12. "The stable isotopes are powerful tracers..."

L15. "that xylem water isotopic compositions effectively reflect source water isotopic compositions." Same goes for the following sentence.

L17. "In this study,..."

L17-19. A biweekly temporal resolution does not qualify as "Highly resolved" (L16). It seems, however, that it is what the authors mean here.

L19. "Using a Bayesian isotope mixing model (MixSIAR), we then quantified the relative contribution to root water uptake of ...". It is important to emphasize the word "relative".

L23. "Xylem water could always be interpreted as a mixture of deep and shallow soil waters from $\delta$18O data, but the $\delta$2H of xylem water was often lower than any other possible water source.". (1) Some isotopic composition value cannot be conceptualized as a water mixture and (2) cannot be depleted/enriched.

L25 and 27. "$\delta$2H decrease" (or analogous) and not "depletion".

L26. "we found that the localization of plant-water source..."

L43-60. There are other assumptions made for determination of relative contributions to RWU across potential sources than the authors H1 and H2, e.g., no sap tissue capacitance, perfect mixing in the sap and no xylem-phloem exchange at the output of the root system, etc. You say this, but the reader has to wait until the the discussion of your results.

L46-47. "The water isotope tracing methodology is commonly used to study plant water uptake...". Here you may cite our review paper on the subject (Rothfuss and Javaux, 2017). It better illustrates your point than the study of Dawson et al. (2002) (which is not in the reference list) as it focuses on RWU solely.

L52. Why "However"?

L54. "the temporal variability in rainfall water isotopic compositions..."

L56-57. "or the relatively higher isotopic composition in fog compared to rain water"

L57-60. Not only. Laser-based spectrometers only allow for a retrospective (off-line) assessment of RWU. Do not forget the development of non-destructive methods for determination of $\delta$soil and $\delta$RWU (Volkmann and Weiler, 2014;Oerter et al., 2016;Gaj et al., 2015;Rothfuss et al., 2013;Volkmann et al., 2016).

L59. "hydrogen and oxygen isotope".

L81. You take too much liberty with terminology. How could "xylem water isotopes" match some "source"? You could write for instance "Plant water source studies in which the xylem water isotopic composition does not spread within the range of the sources' isotopic compositions..."

L83. "hypothesis (H2) is not met"

L86. What does "carry an isotopic signal" mean? I suggest sticking with an "isotopic composition" which is "lower"/"higher"...

L90-91. It reads as if "the isotopic composition of rock water" is "a relevant, alternative plant water source". Please revise grammar.

L101. Do you mean "discrimination increased with decreasing soil water content"?

L104-105. "In fact, a growing number of studies are observing lower xylem water isotopic compositions compared to those of the considered sources"

L108. "The effect of deuterium fractionation on quantification of sources' relative contributions to plant water uptake..."

L109. "hydrogen isotopic composition only..."

L114-121. This belongs to Material and Methods.

L128. What do you mean by "extensive" here? Be more precise in the description of this objective already in the introduction.

L154. Of which two plots?

L155. "In order to measure the xylem water isotopic composition"

L159-160. To "Each soil core was split into top soil (0-10 cm) and deep soil (from 70-80 to 110-120 cm depending on the depth of the rocky layer): even though you underline in the introduction the importance of identifying all water sources (i.e., hypothesis H2), you deliberately omit to sample the soil between 10 and 70(110) cm.

[Figure]

L178-179. I suggest to move this at the end of the §(L148).

L185. I propose: "the pressure in the extraction line..."

L208. Which "significant difference"?

L209-214. A "mismatch" is not precise enough. You may write that you want to "assess if the isotopic compositions of the xylem water samples fall onto the evaporation line in a dual isotopic space". Also I would only present the concept of SW-Excess you are introducing and say that it is an adaptation of the LC-Excess.

L211-212.To "However, because the source water for a tree is more likely to be made of soil water than rain water directly": it is a strange thing to say that trees would directly extract rain water just "less likely" than soil water.

L215. $\sigma$ and $\Lambda$ does not seem appropriate symbols, e.g., $\sigma$ normally stands for standard error. I suggest other symbols like "c" and "d" or "A" and "B", "a'" and "b'"etc.

L218. To "$\delta$2H separation". Since by "$\delta$2H", you mean a numerical value, you shouldn't use "separation", rather "offset", "difference" etc.

L223-227. Why "models" (plural)? MixSIAR is one single model that has different scripts interacting with each other. To "Models were ran in the script version of the package": what other versions of MixSIAR are available? To "the number of Markov chain Monte-Carlo iterations was increased until convergence was reached": did you do this yourself, or was the number of runs optimized automatically? To "Gelman and Geweke diagnostics": what are these? To "residual error term in the isotope mixing models": what is this?

L228. Please define "top" and "deep soil water".

L231-241. You only test the sensitivity of the isotopic mixing model to the definition of the "product", meaning the sap xylem isotopic composition. What about the definition of the "sources"? I am afraid that, since you do not fully characterize the ensemble of

potential water sources, you cannot assess their contributions to tree RWU (see my general comment).

L257. To "3.2. Stable isotopes of tree water sources": do the tree water sources have distinct "stable isotopes" than other sources:)? You might write something like "Stable isotopic composition distribution across tree water sources". Same goes for 3.3.

L258-259. Belongs to M&M.

L266. A slope value of 9.99? Is this a typo? If not, what is the p-value of the linear regression?

L275. "in both hydrogen and oxygen heavier isotopes"

L277. "The isotopic composition of rock moisture". In general, I suggest that you avoid use of "signal" throughout the MS.

L281. No comma in "...budburst), was..."

L284. "had a lower $\delta$2H"

L289. "F. sylvatica presented higher $\delta$18O values (P < 0.05)". Same goes for the rest of the MS: a value cannot be enriched/depleted.

L305-333. You are using the same Bayesian inference model with different parameterization, not a series of different mixing models. This might be confusing to the readers.

L338. "to identify plant water sources and quantify their relative contributions to tree RWU..."

L338-341. I propose something like "lower xylem water isotopic composition than any combination of those of the identified water sources would give"

L341-342. To "The diversity of methodologies used for the extraction of waters and their isotopic determination in all these studies, including ours, rules out potential analytical bias.". Why could these methods not all be biased? Orlowski et al. (2016a) and

Orlowski et al. (2016b) have shown discrepancies between the different methods for water extraction.

L343 and after. If you mean by "offset" the "SW-Excess", then say so. Otherwise, define each time what are these offsets (e.g., is offset the same as in the study of Evaristo et al., 2017?)

L344. "Sternberg"

L380. How do you define a "similar offset" for $\delta$2H and $\delta$18O? Stating that both are linearly linked (e.g., $\delta$2H = a*$\delta$18O + b), would it mean that $\Delta\delta$2H is similar as $\Delta\delta$18O if $\Delta\delta$2H = a*$\Delta\delta$18O ??

L416. Rothfuss and Javaux (2016) is the discussion paper, not the actual article (Rothfuss and Javaux, 2017), please edit this.

L419. Define "SWexcessx" (i.e., already in the M&M).

L422-424. Using both $\delta$18O and $\delta$2H values does not mean you work in a bi-dimensional space. This is strictly speaking a 2x1D space since $\delta$18O and $\delta$2H across water sources are linearly linked. To be able to assess relative contributions to RWU in a 2D space, you need to break this linear bond (via isotopic labeling; e.g., Beyer et al., 2016), so that these water sources spread on some area rather than on some line in a ($\delta$18H, $\delta$2H) coordinate system.

L427. "deuterium fractionation"

References

Beyer, M., Koeniger, P., Gaj, M., Hamutoko, J. T., Wanke, H., and Himmelsbach, T.: A deuterium-based labeling technique for the investigation of rooting depths, water uptake dynamics and unsaturated zone water transport in semiarid environments, J. Hydrol., 533, 627-643, 10.1016/j.jhydrol.2015.12.037, 2016.

Gaj, M., Beyer, M., Koeniger, P., Wanke, H., Hamutoko, J., and Himmelsbach, T.: In-situ

unsaturated zone stable water isotope (2H and 18O) measurements in semi-arid environments using tunable off-axis integrated cavity output spectroscopy, hydrol. Earth Syst. Sci., 20, 715-731, 10.5194/hess-20-715-2016, 2015.

Oerter, E. J., Perelet, A., Pardyjak, E., and Bowen, G.: Membrane inlet laser spectroscopy to measure H and O stable isotope compositions of soil and sediment pore water with high sample throughput, Rapid Commun Mass Spectrom, 10.1002/rcm.7768, 2016.

Orlowski, N., Breuer, L., and McDonnell, J. J.: Critical issues with cryogenic extraction of soil water for stable isotope analysis, Ecohydrology, 9, 3-10, 10.1002/eco.1722, 2016a.

Orlowski, N., Pratt, D. L., and McDonnell, J. J.: Intercomparison of soil pore water extraction methods for stable isotope analysis, Hydrol. Process., 30, 3434-3449, 10.1002/hyp.10870, 2016b.

Rothfuss, Y., Vereecken, H., and Brüggemann, N.: Monitoring water stable isotopic composition in soils using gas-permeable tubing and infrared laser absorption spectroscopy, Water Resour. Res., 49, 1-9, 10.1002/wrcr.20311, 2013.

Rothfuss, Y., and Javaux, M.: Reviews and syntheses: Isotopic approaches to quantify root water uptake: a review and comparison of methods, Biogeosciences, 14, 2199-2224, 10.5194/bg-14-2199-2017, 2017.

Volkmann, T. H., Kühnhammer, K., Herbstritt, B., Gessler, A., and Weiler, M.: A method for in situ monitoring of the isotope composition of tree xylem water using laser spectroscopy, Plant Cell Environ, 10.1111/pce.12725, 2016.

Volkmann, T. H. M., and Weiler, M.: Continual in situ monitoring of pore water stable isotopes in the subsurface, Hydrol. Earth Syst. Sc., 18, 1819-1833, 10.5194/hess-18-1819-2014, 2014.

---

## Author Comment (AC1) · 11 Oct 2018

**Response to referee (Matthias Sprenger)**

**General comments**

Barbeta et al. present a very interesting and extensive stable isotope data set studying the ecohydrology of riparian forests in SW France. Their experimental design of fortnightly sampling over an entire vegetation period is very timely and has pioneer character as most studies addressing similar research questions were limited to few sampling campaigns.

They use well-established methods, both for data gathering (e.g., cryogenic extraction) and data (e.g., MixSIAR) and assess both critically. Their findings regarding a potential deuterium offset and its consequences for plant water uptake studies is timely and will be of interest to a wide range of scientists working with stable isotopes to investigate soil-plant interactions. Their study therefore fits to the scope of HESS and suggest publication after a minor revision addressing the issues raised below.

**We appreciate that the referee found our study pioneering and timely as well as suitable to HESS readership. In the new version of the manuscript, we have now incorporated all his suggestions that we found very constructive and valuable. We are convinced that the manuscript has significantly improved after this revision. A revised version of the manuscript, with modifications in the text and the figures, will be posted after addressing the comments of the other two referees.**

I am not sure if the title is correct, when talking about "Hydrogen isotope fractionation", as you cannot prove that it is a (natural) fractionation process, as there could be also methodological issues causing a mismatch between potential sources and plant water stable isotopes (as you discuss). Maybe "Deuterium offset affects..." is more appropriate.

**We agree that we do not have an explanation for the observed isotopic depletion of twig water compared to all potential water sources. We could however rule out all the main methodological issues that could have caused such an offset, and also gathered evidence from the literature that this seemed to be a rather common phenomenon, although largely overlooked so far and still unexplained. To address the reviewer's concern that the title was not precise enough, we now propose a new and hopefully more accurate title:**

**"*Unexplained hydrogen isotope offsets complicate the identification and quantification of tree water sources in a riparian forest*".**

There is a need for a more critical assessment of the sw-excess correction of xylem data. You refer to Bowling et al. (2017) in the conclusion (xylem water as a potential result of mixing of enriched top soil water and depleted subsoil water), when discussing alternative explanations for xylem water samples plotting below soil water samples in the dual isotope plot. This should be extended as the presented deuterium offset cannot be explained with the current field experiments. Further, xylem waters can also plot above the soil water samples in the dual isotope due to mixed water uptake of soil water and recent infiltrated isotopically enriched rain water. Correction with the sw-excess would not make sense in this case for example.

We agree that correcting xylem $\delta^2$H with the SW-excess has limitations. In fact, this correction is not proposed here as a definitive solution for cases where depletion of $\delta^2$H is suspected. It is rather used as an exercise to illustrate how sensitive the estimated source contributions are to the isotopic inputs. Bayesian models using $\delta^{18}$O and SW-excess-corrected $\delta^2$H led to water source contributions that compared well with those obtained from models using $\delta^{18}$O alone. Also, the variability in source contributions explained by environmental conditions was similar between the two types of models, and with a higher predictive power than with MixSIAR models using just $\delta^2$H or uncorrected $\delta^2$H and $\delta^{18}$O together. We interpreted these results as further evidence that deuterium offsets are at the origin of the problem and that they need to be accounted for to robustly identify tree water sources. We have now added the following text in the Conclusion to acknowledge this important limitation of the SW-excess correction:

*"In addition, the analysis presented here illustrates how sensitive Bayesian mixing models are to the (still unexplained, but seemingly common) $\delta^2$H depletion of xylem water. Although correcting xylem $\delta^2$H with the SW-excess gave better results than without doing it (Table 1), it has to be acknowledged that this correction has some limitations. Under certain conditions, xylem water $\delta^2$H could become momentarily more enriched than soil water, for instance,* due *to mixed water uptake of soil water and recent infiltrated and isotopically enriched rainwater. In those cases, correcting the xylem $\delta^2$H would be misleading. A better understanding of what causes this isotopic separation between xylem and source water is urgently needed."*

**Specific comments**

L 41: Unclear how a climate refugia gets "buffered".

**The sentence was not correctly worded. It is not the climate refugia that get buffered, but instead, they buffer the impact of climate variability in a particular site. The revised text reads now as it follows:**

*"Such mechanisms may also help understand how climate refugia facilitate the persistence of important biodiversity hotspots (McLaughlin et al., 2017)."*

L 54: "the soil depth reached by infiltrating water" reads as if infiltrating water has some kind of boundary beyond which it cannot further percolate. I suggest to refer to mixing processes of water in the subsurface instead.

**We have now modified this sentence according to the referee's comment.**

*"Processes underlying the variability in source water isotopic composition include the temporal variability in rainfall water isotopes and mixing processes of water in the subsurface (Allison and Hughes, 1983; Brooks et al., 2010), the evaporative enrichment of water in surface soil layers (Allison, 1982; Sprenger et al., 2016), the seasonality of groundwater and rock moisture recharge (Oshun et al., 2015) or the relatively higher concentration of heavy isotopes in fog compared to rain water (Scholl et al., 2011). "*

L 51: To clarify that you refer to sampling and not root accessibility, I suggest to say ". . . and can be sampled" (or similar).

**We added "and can be sampled"**

L 94: However, Oerter et al. (2014) that you quote in L 89 actually showed this fractionation for clay minerals.

**To be precise, Oerter et al. (2014) found that cations adsorbed to clay minerals affect the $\delta^{18}O$ of $CO_2$ equilibrated with soil samples, compared to the $\delta^{18}O$ of deionized water initially used to moist the clays, and that the effect depends on the nature of cations ($Mg^{2+}$ and $Ca^{2+}$ deplete the equilibrated $CO_2$ $\delta^{18}O$, $K^+$ enrich it and $Na^+$ leave it unaltered). They explained their results by hypothesizing that "cations adsorbed to the clay surface form isotopically organized hydration spheres of water around them and thereby sequester these water molecules away from the bulk water" and thus from $CO_2$-$H_2O$ equilibration reactions. Because the study of Brooks et al. (2010) was conducted on soils with high clay contents, we agree that the isotopic effects reported by Oerter et al. could have complicated the interpretation of the results of Brooks et al. We now refer to this possibility in the following sentence:**

*"Oerter et al. (2014) suggested that adsorbed cations to clay minerals create isotopically organized hydration spheres of water around them and thereby sequester these water molecules away from the bulk water. However, even if the majority of the water contained in small pores is adsorbed water that does not interact with the more mobile water (the TWW hypothesis), in summer, when only water in small pores is accessible to the trees, there should be an isotopic match between soil pore and xylem water, unless isotopic fractionation occurs during root uptake. In this context, a recent experiment…"*

L 96: This is unclear. If there is only pore water in small pores, then there cannot be any interaction with mobile water, as it is not present.

**We agree with the referee that if in summer there is only water in small pores, there is not any possibility of interaction between mobile and bound water and, should the TWW hypothesis be valid, such interaction would not even happen in spring although mobile water is more readily available. We think that the sentence, reformulated as mentioned above to refer to Oerter's study, clarifies this point.**

L 145: How is the climate defined?

**We have now added the definition of the climate of the area following the Köppen-Geiger classification:**

*"The studied area has a temperate oceanic climate (Cfb in the Köppen-Geiger classification) with a mean annual temperature of…"*

L 229: How was that tested?

**We are not sure if the referee refers here to the pooling of stream and groundwater or the exclusion of fog and rock water as a potential tree water sources. In the first case, stream and groundwater were not statistically different, so it was not possible to distinguish between them in the mixing-models. It could either mean that they are connected or that they are recharged by the same winter rain events. In the case that we had found a strong contribution of stream/groundwater, our data would not be able to clarify if this water pool accessed by the trees was connected to the stream or not.**

**If the comment referred to the fact that we excluded rock moisture as a potential source, this was decided based on the observation that rock moisture was very enriched compared to xylem water but also soil and stream/ground water. This is now shown in the dual isotope plot (Fig. 2), where rock moisture falls in the upper (enriched) part of the LMWL. Regarding fog, it is not just that we only started collecting any fog in our collector by the end of summer (so obviously it could not be a source in spring**

or early summer), but also, as for rock water, fog had a very enriched signal that was never close to xylem water, neither could be involved in any sort of mixture. This is because a contribution of fog would move xylem water samples upper and to the right of the other potential sources in the dual isotope plot, which is the opposite case of what we observed. This is now clearer in the revised manuscript where we explain our choice of potential water sources only after showing their different isotopic composition for the different sampling campaigns (Fig. 2), i.e, lines 227-231 are now moved to the Results section, at the beginning of section 3.4:

*"The potential tree water sources that we considered were restricted to the top and deep soil water and stream/groundwater. Stream and ground waters were pooled together as they were isotopically indistinguishable (Fig. 2). Fog and rock moisture were not included as potential water sources because their isotopic signatures were very enriched compared to xylem water but also soil and stream/ground water (Fig. 2), so that their contribution would have moved xylem water samples up and to the right of the other potential sources in the dual isotope plot, i.e. the opposite of what we observed. Also fog water could only be collected in enough quantity by the end of the summer, so could not have been a significant source of water in spring or early summer. The first sets of isotopic mixing models were run only for the dominant trees of F. sylvatica and Q. robur using…"*

L 252: How did you get volumetric soil moisture? I only see gravimetric moisture described in the methods.

The volumetric water content of rocks was obtained by estimating that limestone rock had a density of 2.5 g cm$^{-3}$. This is now clarified in the text:

*"Using a rock density of 2.5 g cm$^{-3}$, we estimated the mean volumetric water content of limestone rocks to be around 11.75%, which is comparable to that of deep soil."*

L 253: How, where, and how often was the rock water content measured?

Rock water content was measured in every sampling campaign since the second half of July and until the end of the growing season (November). We weighted the rock samples before and after cryogenic extraction, to calculate gravimetric rock water content. We ensured that all water content was removed by the cryogenic extraction by oven drying the material immediately after the extraction and re-weighting it.

L 263 and following: It is not clear which statistical tests were applied, as they are not specifically mentioned in the methods section.

In most of the cases we used generalized linear mixed models, or just general linear models when there were only fixed factors. Although we already mentioned this in the methods, we have now added the following text to emphasize it:

*"The spatial, temporal, species-specific and size-related statistical comparisons between the isotopic compositions of grouped samples were analysed using linear models, or where plot and date were necessarily set as random factors, we used linear mixed models from the package lme4 in R. For instance, for comparisons between groups across several dates, the date of sampling was set as a random factor."*

Fig. 2: Which rain data is shown? Rain that fell on the given days? Rain that fell between the 14 days previous sampling? If so, are they weighted averages? You mention rock moisture in the caption, but I do not find it in the legend.

**Rain water was collected in every sampling date, so the water in the rain collector represents an averaged value of the water precipitated since the previous sampling. The collection was not done based on rain events. We have now clarified this in the revised manuscript, at the end of section 2.1:**

*"Each rain and fog water sample corresponds to the average (amount-weighted) value of the water that precipitated since the previous sampling date."*

**As correctly pointed out by the referee, rock moisture was not shown in Fig. 2, but we have now added it for consistency.**

L 266: Not clear if the SWL was calculated for only topsoil or all soil samples.

**Yes, both top and deep soil samples were used, see L216-217:**

*"The slope and intercept were computed by performing a linear regression on all the soil water isotope data from the surface and deep horizons collected at a given plot and date"*

L 269: Negative effect means more depleted in heavy isotopes with higher antecedent rainfall? Does the isotopic compositions of the antecedent rainfall correlate with the top soil isotopic compositions?

**Yes, the higher the amount of rain fell during the period of collection (ca. 15 days), the more depleted was the top soil water composition.**

L 272: This explanation is not clear to me.

**We thank the reviewer for pointing this out. In fact, there was a confusion in the interpretation of the results, since it is top soil $\delta^{18}O$ that is negatively correlated with top soil water content, not $\delta^2H$. We have modified the sentence and the interpretation of the results accordingly:**

*"In the top soil, $\delta^{18}O$ was significantly (P < 0.05) and negatively correlated with soil water content, but not $\delta^2H$. This is surprising because isotopic fractionation occurring during soil water evaporation and water vapour and liquid diffusion should affect both water isotope signals in the same direction. The fact that these water signals respond differently to top soil water content but similarly to rainfall amount (see above) indicates that observed changes in top soil water isotope signals are primarily governed by precipitation input rather than soil water evaporative enrichment. It may also be that hydrogen isotope of soil water are reflecting extra fractionation processes (e.g. root uptake) compared to their oxygen isotope counterparts."*

L 276: I do not see the rock moisture isotope data in Fig. 2.

**We have now included the rock moisture samples in this plot.**

L 282: Please reword, as "evaporation line" is evaporation line is defined as the linear change in the isotopic composition in dual isotope space of a single water source through evaporative concentration of the remaining liquid. However, these xylem samples were taken all at the same time. I do not question that they underwent evaporation and kinetic fractionation, but "evaporation line" is the wrong word.

**We agree with the reviewer that "evaporation line" is not the correct term here. We have modified this sentence:**

*"Xylem water from the first campaign in late April (i.e. just before or during budburst), was exceptionally enriched (Fig. 3) and fell in the upper right part of the dual-isotope space (green "outliers" on the right side of Fig. 2)."*

L 286: Needs rewording, as "had a negative effect" implies that actually DBH influences the xylem water isotope compositions. I suggest saying something like "more depleted with higher DBH".

**We have replaced "had a negative effect on" by "were negatively correlated with".**

L 291: This questions that grouping xylem and root samples in Fig. 2, Fig. 3, and Fig. 5. You should then present each data set separately there already. Or do you not present any extracted root water there? Not clear to me.

**Water extracted from above-ground coarse roots is also considered as xylem water. The $\delta^2H$ offset and their spatio-temporal patterns still persists after excluding the water extracted from (outcropping) coarse roots. In order to ensure that those trees did not drive all the statistical results, we selected "plot" as a random factor, to control for any effects related to sampling (root xylem samples were all from the same plot). We now clarified this point:**

*"The four trees (all on the same plot) in which xylem water was extracted from outcropping coarse roots (rather than from twigs) showed a significantly more depleted $\delta^2H$ over the whole season (P < 0.001), but no significant difference in $\delta^{18}O$, compared to all the other trees (Fig. 4). The $\delta^2H$ offset still persisted after excluding these coarse root samples, demonstrating that xylem water $\delta^2H$ exhibited different patterns than $\delta^{18}O$."*

L 306: What does "only for the dominant trees" mean? Why not all sampled trees?

**We only sampled non-dominant trees for *F. sylvatica*, not for *Q. robur*. Consequently, we preferred to compare only the water sources of dominant trees for this species comparison. This is now clarified:**

*"The first set of isotopic mixing models were run only for the dominant trees of F. sylvatica and Q. robur using both $\delta^{18}O$ and $\delta^2H$ data. Because non-dominant trees were only sampled for F. sylvatica, not for Q. robur, we preferred to exclude them when comparing the two species."*

Table S3: I encourage the authors to add this Table S3 to the main manuscript, as this presents a major finding of the study. Why did you decide to take 5-day averages and amounts?

**We agree with the referee. We now moved Table S3 to the main text. We chose 5-day averages and amounts because they gave the strongest correlations with source contribution overall (better marginal r-squared).**

L 354: The "water isotope fractionation occurring at the soil-root interface" would not cause the deuterium offset, as Vargas et al. (2017) describe it as a Raleigh type fractionation.

**We do not understand this comment. Vargas et al. do report a deuterium offset between xylem and soil water, with more depleted xylem samples compared to soil water. They conclude that this is caused by fractionation during root water uptake. They then elaborate on the effect of such a fractionation on**

the remaining soil water pools, and conclude that this should cause a progressive enrichment of soil water via Rayleigh distillation effects, at least when soil water is scarce (i.e. in summer). Such mechanisms would completely support our results. In any case, here, we do not refer to one specific mechanism, but rather to an array of mechanisms that can occur in the soil-root interface, including those suggested by Vargas et al (2017).

L 360: You mention a carbonate rich C Horizon in the methods. Could processes described by Meißner et al. (2014) be relevant here?

This is a very good point. Indeed, similar to our study, Meißner et al. had to evoke either positive ($\delta^{18}$O) or negative ($\delta^2$H) offsets between xylem and soil water in an attempt to reconcile both isotopes at identifying a common water source. They also performed extra lab experiments by manipulating clay and carbonate contents to try explain these putative offsets. They report that HCl-treated (i.e. carbonate-free) soil samples have a cryogenically-extracted water $\delta^{18}$O in agreement with that of input water whereas the $\delta^{18}$O of cryogenically-extracted water from carbonate-rich soil samples is depleted compared to input water. On the other hand, they find no effect of carbonate content on deuterium isotopes. They thus conclude that "H isotopes probably reflect the plant water uptake […] whereas O isotopes in extracted water are shifted to lower values as compared to [soil water and thus] plant water uptake". They suggest that the $\delta^{18}$O depletion of extracted water was caused by temperature-dependent oxygen isotope exchanges between soil water and carbonates during the extraction, but they did not perform tests at different extraction temperatures to verify this was the case. Also the carbonate-induced O isotope effect that they report is about -1‰, which is only half of what would be required to fully reconcile their $\delta^{18}$O and $\delta^2$H data from the field. In our case, the presence of carbonates in the C horizon would be responsible for a $\delta^{18}$O depletion of extracted water from the deep soil samples of about 1‰, i.e., the "true" soil water in this horizon should be shifted by about +1‰. This would slightly modify the SW-excess values but would not cancel the observed isotopic offset between soil water and xylem water. Therefore, the results of Meißner et al. (2014) are relevant, but do not explain the isotopic offset observed here. This is now stated in the discussion of the revised manuscript, in the second paragraph of section 4.1.

*"Another possibility is that fractionation processes occur during water extraction. Meißner et al. (2014) reported that treating soil samples with HCl to remove carbonates prior to water extraction led to a cryogenically-extracted water $\delta^{18}$O in agreement with that of input water, whereas the $\delta^{18}$O of cryogenically-extracted water from carbonate-rich soil samples was depleted by about 1‰ compared to input water. On the other hand they found no effect of carbonate content on deuterium isotopes. They suggested that the $\delta^{18}$O depletion of extracted water was caused by oxygen isotope exchanges between soil water and carbonates during the extraction, a process that should be temperature-dependent. Oerter et al. do not specify their extraction temperature but we expect it to be > 60°C, i.e. close to our extraction temperature of 80°C, so that we could expect a carbonate-induced isotope effect of comparable magnitude. However, if the presence of carbonates in the C horizon would be responsible for a $\delta^{18}$O depletion of extracted water from the deep soil samples of about 1‰, this would mean that the "true" soil water in this horizon should be shifted by about +1‰. This would slightly modify the SW-excess values but would not cancel the observed isotopic offset between soil water and xylem water. Therefore, although the results of Meißner et al. (2014) are very relevant to our study, they cannot explain the isotopic offset observed here."*

L 362: I would disagree with this generalization of more tightly bound water being generally more depleted, as it seems to be depending on the time of sampling: During summer, mobile water will be more enriched in heavy isotopes as the rainfall is relatively enriched, as the more tightly bound water, which is likely to have a isotopic composition that stems from winter (more depleted) rainfall/snowmelt. I discussed this issue in Sprenger et al. (2018). However, as most field sampling takes place during summer, various studies showed more isotopically depleted waters in more tightly bound water than mobile water (Geris et al., 2015; Oerter and Bowen, 2017).

**We agree that, in the conceptual framework of the TWW hypothesis, water in small pores (also referred sometimes to "bound" water, because assumed locked away from the more mobile water flow) is not necessarily more depleted, as it may come from isotopically-enriched rain events (e.g. during summer storms…). This said, in this part of the text we are not referring to "bound" water in the sense used by Brooks et al (2010), to cite an example. Note that we do not use in fact the term "bound", but the term "adsorbed". This is because we wanted to make a distinction between (1) bound water *sensu* Brooks et al. (2010), which means water that does not flow and remains in soil micropores and (2) adsorbed water, which is the physically adsorbed water on mineral or organic surfaces. This adsorbed water is held at higher tensions and should thus be less accessible by plants but is not necessarily disconnected from mobile water and can still exchange isotopically. This adsorbed water, at equilibrium, is expected to always be more depleted than mobile water (Lin et al. 2018; Lin & Horita 2016). Note however that we found clear depleted values of xylem water with respect to bulk soil water during the wetter periods, i.e., when the proportion of absorbed water relative to bulk soil water should be minimal and thus contribute little to plant water use. This seems to indicate that the isotope arrangement between adsorbed and mobile water within soil pores does not seem to be responsible for the hydrogen isotope offsets reported here.**

L 375: Why "weak spatiotemporal variability"? The variability is partly very high (see F. sylvatica on July 4). It's the pattern that is weak, which the current study cannot really well explain.

**We agree that the term weak used to characterize the spatiotemporal variability in SW-excess was inaccurate, and unnecessary for the argument. We removed it from the text:**

*"… given the spatiotemporal variability in SW-excess (Fig. 5)".*

L 385: In the light of Zhao et al. (2016) and your discussion in L 390 and following, do you then think that cryogenic extraction is then the best way to sample xylem water? Consider discussing potential impacts of not limiting the sampling to the actual transpiration water when using cryogenic extraction. However, why is SW-excess then so variable?

**Cryogenic extraction retrieves all the water in the stem (after removing bark and phloem). Plant water storage pools connected to xylem vessels through a fractionating symplastic pathway with its own temporal dynamics could be behind the observed isotopic offset between bulk stem water and source water. It is technically challenging to design a non-fractionating extraction technique to separate these water pools in the stem. Vapor equilibration or probe-based sampling may produce similar results, but it remains to be tested. A possible explanation for the variability in the SW-excess is that the proportion of each of these water pools varies over time. We have reformulated slightly the end of the discussion on this topic (end of section 4.1):**

*"For the species in this study, ray and axial parenchyma can account for around 31% of total xylem tissue volume in both F. sylvatica and Q. robur (Morris et al., 2016). Storage water in the stem can account for up to 16% of daily transpiration in F. sylvatica (Köcher et al., 2013), and contribute even more in some subtropical tree species (Oliva Carrasco et al., 2015). Future studies are now required to explore the role of symplastic water transport and storage as a potential mechanism leading to the depletion of bulk wood water $\delta^2H$ compared to the actual source water signal. This mechanism may be quantitatively relevant for interpreting the isotopic composition of bulk xylem water in terms of source water and explaining the variability in SW-excess reported here."*

L 429: As you pick up the Table S3 again as one of the main findings, I highly suggest including it in the main manuscript and not having it in the supplementary material.

**Following referee's advice, we have now included Table S3 in the main document.**

L 432: This is a very important point you raise. How did you deal with xylem data of positive SW-excess? This could be a result of a mixed source (soil water plus recently infiltrated rain water). I can also imagine a situation, where an enriched soil water source in the top soil with a depleted source (e.g., snowmelt) in deep soil results in a xylem water isotope composition plotting below the sw-line, but does not result from deuterium fractionation. Consider discussing the limitations of your sw-excess correction.

**As explained in the text, in one of the Bayesian mixing models, all xylem $\delta^2$H data were corrected for SW-excess. This means that positive offsets (very rare) were also corrected, although they clearly indicate that they were caused by multiple factors. We slightly modified the end of this discussion to warn about the limitations of the SW-excess:**

*"Based on these correlations, correcting xylem water isotopes using SW-excess appeared to improve the power of the dual-isotope approach. However, systematically correcting xylem data with the SW-excess is also problematic because non-zero SW-excess are not only caused by a hydrogen isotope fractionation between source and xylem waters. Other water sources than soil water could also contribute to the xylem signal, or spatiotemporal dynamics in the soil water isotope profile could also complicate the concept of soil water isotope line and thus SW-excess. The fact that we sometimes observed positive SW-excess indicates that we do not only correct for one single fractionation factor, and demonstrates the limitation of the SW-excess correction proposed here."*

**Technical corrections**

L 18: No delta sign (applies to the entire abstract)

We revised the entire abstract accordingly.

L 18: bi-weekly could mean twice a week or every two weeks (fortnightly); applies also to L 154 and L 259.

**We have replaced bi-weekly by fortnightly throughout the manuscript to avoid the possible confusion.**

L 31: This is not a question. You either need to remove the question mark or rephrase.

**We reformulated the question to:**

*"Why is an improved understanding of tree water use needed?"*

L 39: ecohydrological

**Changed.**

L 41: understanding

**Changed.**

L 46: Somewhere here you need to introduce "stable isotopes of water (2H and 18O)", as you currently miss this, but use 2H and 18O it in L 67 without having it introduced.

**We introduce them before now.**

L 68: No hyphens

**Changed.**

L 69: . . .rainless summer fills small soil pores first and does not contribute to river flow nor to mixing. . .

**Changed.**

L 74: TWW hypothesis holds, . . .

**Changed.**

L 78: . . .stream water for riverside trees.

**Changed.**

L 96: . . .when only water in small pores is accessible. . .

**Changed.**

L 97: . . . between sampled soil pore and . . . (important to mention "sampled", as this match is solely based on H1 and H2)

**Changed.**

L 122: "to the data sets" appears to be more confusing than clarifying.

**Deleted.**

L 146: 813 mm/year (no digit needed and the unit is mm/year)

**Changed to 813 mm yr$^{-1}$.**

L 191: composition (given in delta notation . . .) .

**Changed.**

L 214: I suggest to not use sigma, but a different letter, since it could be misinterpreted as standard deviation.

**We changed σ to Γ to avoid any confusion.**

L 224: were run

**Corrected.**

Fig. 1: It seems that there is something wrong with the x-axis. The VPD data is shown as a continuous measurement, but the x-axis is not really a axis, as the x-axis ticks are not equally distributed (e.g., every 14 days). Not sure if the date format is correct according to the Copernicus guidelines, and it's actually different format to Fig. 2. The streamflow data does not look like actual data per second, but seems to be an average value for the individual days. If so, this should be mentioned. Add mm/day for precipitation. Thus, GWC should have a different x-axis as the other three subplots.

**The referee is right regarding the coherence of the x-axis. We previously decided to simplify the figure by using a common x-axis. We have now redrawn the figure, with the correct and common time axis for all variable. They all belong to the same period, but there is daily (VPD and rainfall), monthly (streamflow) and fortnightly (GWC) data. We corrected the units on the y-axis and the figure legend accordingly.**

L 260: I suggest defining delta notation in the methods and then you do not need to refer to VSMOW-SLAP scale here.

**We agree that the isotopic scale should be stated in the Methods section. We included a sentence there (at the end of section 2.2):**

**"*All isotopic data reported here are expressed on the VSMOW-SLAP scale.*"**

**The reference to VSMOW-SLAP on l. 260 was then deleted. On the other hand, we kept the scale on all the figures to make sure they are fully comprehensive on their own.**

Fig. 2: I encourage the authors to consider using in addition to different colors also different markers for e.g., soils, vegetation, precipitation… just to make it easier to see the differences in the dual isotope plots. Please also consider increasing the marker size. I suggest defining delta notation in the methods and then you do not need to refer to VSMOW-SLAP scale here. Typo: ". . .soil at two. . ." and "The blue line corresponds. . .black line corresponds to. . ."

**We have re-drawn this figure, in order to include rock and fog water for the campaigns that those were available. We also increased the marker size to improve the visualization, but preferred not to use different marker types given the large number of categories. The high-resolution figure published in the final version of the manuscript should be good enough to distinguish the already contrasting colors.**

Fig. 3: Consider adding "cm" to 50-120. I suggest defining delta notation in the methods and then you do not need to refer to VSMOW-SLAP scale here. Not sure if the date format is correct according to the Copernicus guidelines, and it's actually different format to Fig. 2.

**We added cm in the figure legend as proposed but kept reference to the VSMOW-SLAP scale for reasons explained above.**

Fig. 4: Please use different colors, as this is misleading when similar colors are used in Fig. 3 even though there is different data shown (soil vs roots). I suggest defining delta notation in the methods and then you do not need to refer to VSMOW-SLAP scale here.

**The brown color used here for roots is exclusive to this type of sample/plant organ. However, we agree that it may be hard to distinguish from the brown color used to indicate top soil water in the other figures. We thus decided to use only green colors for this figure and to distinguish between light (*F.**

*sylvatica*) and dark (*Q. robur*) greens as in the other figures. The x-axis is clear enough to separate twig and root samples without a color scheme.

Fig. 6: Why hyphenated plant-water?

**We have removed the hyphen.**

**References**

Craig, H., Gordon, L. I. and Horibe, Y.: Isotopic exchange effects in the evaporation of water, J. Geophys. Res., 68(17), 5079–5087, doi:10.1029/JZ068i017p05079, 1963.

Lin, Y. and Horita, J.: An experimental study on isotope fractionation in a mesoporous silica-water system with implications for vadose-zone hydrology, Geochim. Cosmochim. Acta, 184, 257–271, doi:10.1016/j.gca.2016.04.029, 2016.

Lin, Y., Horita, J. and Abe, O.: Adsorption isotope effects of water on mesoporous silica and alumina with implications for the land-vegetation-atmosphere system, Geochim. Cosmochim. Acta, 223, 520–536, doi:10.1016/j.gca.2017.12.021, 2018.

McLaughlin, B. C., Ackerly, D. D., Klos, P. Z., Natali, J., Dawson, T. E. and Thompson, S. E.: Hydrologic refugia, plants, and climate change, Glob. Chang. Biol., 23(8), 2941–2961, doi:10.1111/gcb.13629, 2017.

Sprenger, M., Leistert, H., Gimbel, K. and Weiler, M.: Illuminating hydrological processes at the soil-vegetation-atmosphere interface with water stable isotopes, Rev. Geophys., 54(3), 674–704, doi:10.1002/2015RG000515, 2016.

---

## Author Comment (AC3) · 12 Nov 2018

**Response to referee (Xin Song)**

**General comments**

In this study, Barbeta et al. applied stable isotope techniques to investigate potential water sources of two broad-leaved tree species in a temperate, riparian forest. For this purpose, they made collection of a season-long dataset of d18O and dD compositions from tree xylem water, soil water at different depths, and other potential water sources. They show for both of the tree species that different water sources can be appropriately identified with oxygen isotope data. However, the same conclusion cannot be drawn from hydrogen isotopes, as dD of xylem water on many occasions apparently fell out of the range as encompassed by the potential sources. The authors made a detailed discussion of several possible causes of the observed xylem-soil water dD mismatch, and concluded that isotopic fractionation in the unsaturated zone and/or within plant tissues could well be the driving mechanism.

This is a well-designed field study that addresses an important topic in stable isotope ecohydrology. The manuscript is well written, and data analysis was carried out in a solid manner. The finding of dD fractionation in two commonly occurred temperate tree species is a timely reminder that this phenomenon may not be an exception as previously thought, only restricted to a narrow range of species such as mangroves or species from arid regions, but more likely one that is common to a wider range of species. This finding will have important implications for stable isotope ecohydrological and ecophysiological studdies.

**We very much thank the referee for his positive assessment of the manuscript. We have amended the manuscript following his comments. Please find below the responses to the referee's comments.**

**Specific comments**

1. Line 232: There is a lack of explanation of the rationale behind the modeling exercise of using SW-excess corrected dD. As far as I understand, given the possibility that fractionation occurs at the soil-root interface or within plant tissues, the purpose of applying such type of correction is to obtain real dD values of the water that is available for uptake by plant roots (i.e. correcting dD back to the point before fractionation occurs). If this is the case, then an a priori assumption for doing correction based on a SW-excess line would be that soil water pools are the only sources of water available for tree roots to take up (otherwise an observed "apparent" SW-excess could have been caused by contribution from non-soil sources that do not necessarily follow the SW-line). Was such an assumption met in the present study? How general applicable is this sw-excess based method to other studies? I think it is worthwhile for the authors to discuss further on these important points (this could be done either in M&M or Discussion).

**The referee is right. The SW-excess correction carries out the assumption that tree water uptake occurs exclusively in soil water pools, although stream or groundwater pools are lumped together and kept in the analysis when applying the mixing model. As explained in the text, we ruled out fog and rock water as potentials sources because their isotopic signal is much more enriched than xylem water (Fig. 2). The assumption of exclusive root access to belowground (soil and stream/ground) waters seems thus reasonable. We were not able to propose a correction that would also include stream/ground water, because stream/ground water sometimes plotted outside (above) the soil water line (Fig. 2). However in a lot of cases, stream/ground water coincided with the lower part of the soil water line, thus justifying**

their inclusion in the mixing models, even after the SW-excess correction on the xylem waters. Also, based on the observation that the dual isotope approach with the corrected $\delta^2$H yields source contributions that had a stronger relationship with environmental conditions (*e.g.* the higher the rainfall amount, the higher the top soil water contribution, Table 1, now in the main document), we concluded that our correction and the underlying assumption was justified. We have now better detailed this rationale in the Methods:

*"Correcting xylem $\delta^2$H with SW-excess implies that tree water uptake relies only on soil water pools because the SW-excess is calculated using the slope and intercept of the soil water line. However, the lower part of this line usually overlaps with unenriched stream/ground water. Thus, we expected that $\delta^2$H departures from this line are meaningful in potential cases where trees are accessing not only soil water but also stream water."*

While adding this remarks in the Discussion:

*"This exercise was made with the purpose of providing a sensitivity analysis. However, its use in other sites and plant species should be made with caution, as it is very likely that the observed $\delta^2$H offset may display different patterns depending on other water sources."*

2. Line 272: I understand the authors' argument for a lack of sensitivity of d18O to soil water content here, but still the line "changes in the isotopic composition of soil water with rain addition…" seems somewhat contradictory to what is already stated in Line 269 "rainfall amount (…) had a negative effect on top soil water d18O…". Isn't it like saying that "rain addition has a significant effect on d18O" versus "rain addition may not cause sufficient changes in d18O"?

This is correct. As we already explained in our answer to referee #1, there was a confusion in the interpretation of the results, since it is top soil $\delta^{18}$O, not $\delta^2$H, that is negatively correlated with top soil water content. We have now modified this sentence and the interpretation of the results accordingly:

*"In the top soil, water content was negatively correlated with $\delta^{18}$O (P < 0.05), but not with $\delta^2$H. This is surprising because isotopic fractionation occurring during soil water evaporation and water vapour and liquid diffusion should affect both water isotope signals in the same direction. The fact that these water signals respond differently to top soil water content but similarly to rainfall amount (see above) indicates that observed changes in top water isotope signals are primarily governed by precipitation input rather than soil water evaporative enrichment. It may also be that hydrogen isotope of soil water are reflecting extra fractionation processes (e.g. root uptake) compared to their oxygen isotope counterparts."*

3. Line 164: How often was stream and groundwater collected? This sentence reads as if they were collected every day? Or on every sampling campaign?

Stream and groundwater collected were collected for every sampling date that we sampled the trees and the soil, so fortnightly. We have slightly modified the sentence to clarify this:

*"we collected water from the stream for every sampling date"*

4. Line 276: where is the rock moisture data in Fig. 2?

Because we did not include rock moisture in the source contribution analysis, we had initially removed it from the plots. Finally, we have decided to include it as it has its importance in illustrating our decision of not considering rock moisture as a potential source (see revised Fig. 2).

5. Lines 280-284: I'd like to argue here that if xylem water had become progressively enriched due to stem evaporative enrichment over the past winter, we would expect xylem water to deviate not only from the LMWL line but also from SWL. Yet, from fig. 2 soil and xylem water appear to fall into pretty much the same line.

**We have not found much literature reporting branch evaporative enrichment. However, in a recent paper (Bowling et al. 2017, see references in the manuscript), a similar early spring stem isotopic enrichment was attributed to water losses prior to budburst. While we cannot be completely sure that this is the case for the studied trees, there is no apparent reason than branch evaporation would not produce a water evaporation line with a similar slope than soil. We have now changed a "was" by a "could be", which is more appropriate in this case:**

*"This could be indicative of stem evaporative enrichment over winter."*

6. Line 346: change "have also reported isotopic offsets" to "have also been reported to display isotopic offsets"

**Changed.**

7. Lines 360-361: Is there also a possibility that cryogenically extracted soil water does not truly represent bulk soil water? See a recent Ecohydrology paper by Orlowski et al. (2016) Critical issues with cryogenic extraction of soil water for stable isotope analysis

**Yes, this a possibility that we considered. However, internal tests of our cryogenic extraction line showed non-significant discrepancies between spiked and extracted water in sands. In fact, sandy soils are much less affected by methodological issues of cryogenic extraction. Besides, we would expect effects on both water isotopes, not only on $\delta^2$H data.**

8. Line 379: change the second comma to semicolon

**Instead of doing this, we have added a period:**

*"Similarly, we do not think that branch evaporation is responsible for the reported isotopic offset (Martín-Gómez et al., 2017). If it were the case, we would expect the magnitude of the offset to vary over the season with evaporative demand and to affect both hydrogen and oxygen isotopes, i.e., the opposite of what we report here."*

9. Line 381: "fore example" should be "for example"

**Changed.**

10. Line 387: "sympastic" should be "symplastic"

**Changed.**

11. Line 444: this is not a complete sentence

**We have amended this sentence.**

12. Line 445-446: The idea is great, but may not be easy to realize with the current extraction based method that is only capable of extracting bulk water from a plant tissue. Methodological advancement is apparently needed to confer ability of separately analyzing water from different pools (i.e., parenchyma cells versus xylem water) within a given plant sample.

**We agree with the referee. We have thus added the following sentence:**

*"Secondly, to obtain a better understanding of the isotopic dynamics of water pools within plant tissues, notably those associated with plant storage water and its dynamics (Pfautsch et al., 2015) which will require developing new extraction methods for xylem water".*

13. Line 448: change "fraction" to "fractionation".

**Changed.**

14. Fig. 5: minus signs are missing in several places of the y-axis.

**We have corrected this.**

---

## Author Comment (AC4) · 12 Nov 2018

**Response to referee (Youri Rothfuss)**

**General comments**

In their manuscript submitted to HESS and currently in its discussion phase, Barbeta and colleagues investigate the water sources of *F. sylvatica* and *Quercus robur* (L.) on basis of water stable isotopic measurements in sap xylem, stream/groundwater, and soil water in a vertical profile. For this, they opt for the most commonly used method, which is inversing the isotopic data with statistical modeling. They choose MixSIAR, a mixing model embedded in a Bayesian framework. A particularity of this study is that xylem water isotopic samples of both species very often not plot on the soil evaporation line in a ($\delta^{18}$O, $\delta^2$H) coordinate system, which is explained by the authors as being due to a significantly higher hydrogen than oxygen isotopic fractionation during root water uptake. The manuscript is generally well written (aside sometimes from the isotopic terminology – see my specific comments), easy to follow, and falls into the scientific scope of HESS. My general concern is that there is a major contradiction between the first part of hypothesis H2 ("it is essential that all potential water sources are identified and accessible") and the fact that the authors deliberately do not sample from the soil between 10 and 70 (or 110, "depending on the depth of the rocky layer") cm depth. In a review article (Rothfuss and Javaux, 2017), a simple synthetic experiment showed how much the non-fulfillment of H2 could lead to biased estimates of the relative contributions to plant root water uptake across a set of potential sources. If the authors can provide proof that the "mid-soil" is isotopically not any different than another source (e.g., deep soil), then it can be pooled together with this other source (e.g., "mid-deep soil"). If this is not the case, the authors' analysis might not be valid. For instance, we cannot say for sure that the observed hydrogen isotopic fractionation is not in fact partly due to the values of isotopic compositions in the missing soil water source.

**We appreciate that the referee found our study well written, easy to follow and falling into the scope of HESS. We have followed his comments and suggestions to improve the general quality of the manuscript, notably regarding the "isotopic terminology". We have also addressed the concerns of the referee in relation to our sampling strategy and associated analysis. We fully agree that the identification and sampling of all possible water sources (H2) needs to be fulfilled in order to achieve realistic estimations of water source contributions. However, we have several lines of evidence that this was the case in our study, as explained below.**

**Our sample strategy of soil water samples was designed to capture the spatial variability of soil water isotopic composition as much as possible. We selected three plots that differed in their microclimatic and topographic conditions (that in the end did not imply differences in tree water uptake), and in each of these plots, we sampled three subplots. This allowed capturing both the plot- and tree-scale variability in soil water isotopes. We expected that the temporal and spatial variability of soil water isotopes would be large in the top soil and small in the deep soil, because of the spatially heterogeneous effect of soil evaporative enrichment that will vary with tree cover, sunflecks frequency, throughfall, stemflow, windspeed… The extent of evaporative enrichment of soil water isotopes is commonly limited to the top 20-30 cm of the soil (Sprenger et al. 2016). Thus, our sampling strategy of collecting only top (10cm) and deep (70-110cm) soil water was an efficient way to capture the two ends of the evaporative enrichment soil water isotope profile.**

We recognize that soil water evaporative enrichment can sometimes extend to deeper soil layers (e.g. 50 cm) but only under prolonged periods of high evaporative demand and low soil water contents, typical of Mediterranean or semi-arid regions (Allison et al., 1983). In these situations, the evaporative front is usually located below the soil surface and the isotope profile displays a maximum value at the location of the front rather than at the soil surface. However, such situations are unlikely to occur at our studied site, especially during the year of the sampling (2017) that was dry only very early in the season when evaporative demand was low (as noted in the manuscript, the permanent wilting point was reached in the top soil only at one date in September).

Our expectation of low spatio-temporal variability of soil water isotopes in the deep layers (70-110cm) was confirmed by our data (Figs. 2 and 3), indicating that most of the variability in soil water isotopes came from changes in evaporative enrichment and recharge of the top soil layer by summer rainfall. We also noted that the deep soil and top soil isotopic composition often plotted in the dual isotope space on an evaporation line (Fig. 2), implying that the water isotope composition of middle soil layers must necessarily fall in between. Only when the sampling date followed abundant rainfall (e.g. July 4th, Fig. 2), the soil water line was somewhat altered and the top soil layer displayed a water isotope composition close to rain water and more depleted than deeper in the soil profile, i.e., the opposite of an evaporative front. We already reported in the manuscript that the estimation of source contributions in such situations may be misleading (Fig. 8).

Our conclusions are also confirmed by a more intensive sampling campaign performed at one date in late summer 2018 where soil samples were collected every 15-20cm from soil surface to below 70cm (Fig. S1, shown below), The water isotope profile in the soil from this campaign displays a typical evaporative enrichment profile with more depleted values at depth and no statistical differences among the soil layers below 35 cm for both water isotopes (Fig. S1). This reinforces our argument that below 35cm, the isotopic composition is fairly stable, and sometimes very similar to that of groundwater and stream water (Fig. 2). Although this "test" sampling was only done for one date in late summer when evaporation had time to shape this isotopic profile in depth due to dry conditions, we think that this was the most common case for our sampling campaigns in 2017, given the clear soil evaporation lines we could draw for each date (Fig. 2).

To sum up, our data seems to support the appropriateness of grouping several deep layers into one "deep soil" water pool, which was also considered the better option for balancing our sample processing capacity and the information provided by the data. In addition, we find it highly unlikely that the depleted $\delta^2$H of xylem water is caused by water uptake from this middle layer. This would require a pervasive (over time and space) negative peak in $\delta^2$H in this middle layer. We cannot think about any ecohydrological or physico-chemical process driving a -8‰ depletion in only one of the isotopes and in a targeted soil layer, without affecting the surrounding layers.

[Figure]

**Fig. 1.** $\delta^{18}O$ and $\delta^2H$ at different soil depths. Different letters indicate significant differences between depths (*P*<0.05).

**Specific comments**

L12. "The stable isotopes are powerful tracers. . ."

**We have slightly modified the Abstract and omitted this sentence.**

L15. "that xylem water isotopic compositions effectively reflect source water isotopic compositions." Same goes for the following sentence.

**The first sentence was changed to "that the stable isotope composition of xylem water effectively reflects that of source water".**

**The following sentence was changed to "However, this assumption has been called into question by recent studies that found that, at least at some dates during the growing season, plant water did not reflect any mixture of the potential water sources."**

L17. "In this study,. . ."

**Changed.**

L17-19. A biweekly temporal resolution does not qualify as "Highly resolved" (L16). It seems, however, that it is what the authors mean here.

We meant exactly what is said, i.e., that "Highly resolved datasets covering a range of environmental conditions could shed light on possible plant-soil fractionations processes." Techniques monitoring continuously (i.e. at a hourly time step) the stable isotope composition of xylem and soil water are currently under development and already providing interesting results, as mentioned by the referee below. Unfortunately, to the best of our knowledge, such methods have not yet been implemented to report data for extended periods of time (*e.g.* a full growing season) and thus over "a range of environmental conditions". This is the latter aspect that we tried to cover with our sampling strategy.

L19. "Using a Bayesian isotope mixing model (MixSIAR), we then quantified the relative contribution to root water uptake of . . .". It is important to emphasize the word "relative".

**Changed.**

L23. "Xylem water could always be interpreted as a mixture of deep and shallow soil waters from δ18O data, but the δ2H of xylem water was often lower than any other possible water source.". (1) Some isotopic composition value cannot be conceptualized as a water mixture and (2) cannot be depleted/enriched.

**1. The sentence does not imply that xylem water can only be a mixture of deep and top soil water (or any other source), but rather that this is what we observed in this particular site and growing season (for $\delta^{18}O$). This is compatible with situations in which xylem water is not a mixture but it is sourced in just one soil layer or any other water pool.**

**2. We believe that the term "depleted" when used to compare the isotopic composition of two water pools is perfectly valid and generally accepted.**

L25 and 27. "δ2H decrease" (or analogous) and not "depletion".

**We have kept the term "depleted" that we consider correct, because it is explicitly said in the previous sentence that this depletion is relative to the other water pools (i.e. "more depleted than any other possible water source"). We felt unnecessary to repeat it again in this sentence, and did not want to use another word (offset, decrease…) that would only bring more confusion.**

L26. "we found that the localization of plant-water source. . ."

**Changed.**

L43-60. There are other assumptions made for determination of relative contributions to RWU across potential sources than the authors H1 and H2, e.g., no sap tissue capacitance, perfect mixing in the sap and no xylem-phloem exchange at the output of the root system, etc. You say this, but the reader has to wait until the discussion of your results.

**We agree. However, since this is not a review paper, we did not want to extend too much on the methodology and its underlying assumptions. We revised assumption H1 to include these other assumptions that the referee mentions without going into much more details:**

**"Firstly (H1), it is assumed that isotopic fractionation during root water uptake (and/or xylem water redistribution) does not occur…"**

L46-47. "The water isotope tracing methodology is commonly used to study plant water uptake. . .". Here you may cite our review paper on the subject (Rothfuss and Javaux, 2017). It better illustrates your point than the study of Dawson et al. (2002) (which is not in the reference list) as it focuses on RWU solely.

**We now cite Rothfuss and Javaux (2017) here and have sorted out our reference list. Thank you for pointing this out!**

L52. Why "However"?

**We have removed it.**

L54. "the temporal variability in rainfall water isotopic compositions. . ."

**Changed to "temporal variability in the isotopic composition of rainfall".**

L56-57. "or the relatively higher isotopic composition in fog compared to rain water"

**Changed to "or isotopic processes during fog water droplet formation", as fog is not necessarily more enriched than rain water at the monthly scale.**

L57-60. Not only. Laser-based spectrometers only allow for a retrospective (off-line) assessment of RWU. Do not forget the development of non-destructive methods for determination of δsoil and δRWU (Volkmann and Weiler, 2014;Oerter et al., 2016;Gaj et al., 2015;Rothfuss et al., 2013;Volkmann et al., 2016).

**We fully agree but again, this is not a review paper. Here we just wanted to emphasize the fact that the development of laser-based spectrometers opened up the possibility to perform more extensive retrospective assessments of RWU, which is fully relevant for our study.**

L59. "hydrogen and oxygen isotope".

**Changed to "hydrogen and oxygen stable isotopes".**

L81. You take too much liberty with terminology. How could "xylem water isotopes" match some "source"? You could write for instance "Plant water source studies in which the xylem water isotopic composition does not spread within the range of the sources' isotopic compositions. . .".

**We appreciate the suggestion of the reviewer and have modified the text accordingly.**

L83. "hypothesis (H2) is not met"

**Changed.**

L86. What does "carry an isotopic signal" mean? I suggest sticking with an "isotopic composition" which is "lower"/"higher". . .

**"Isotopic signal" is commonly used in the isotope geochemistry literature without ambiguity. We nevertheless followed the suggestion of the referee and modified the sentence, that now reads:**

*"Moreover, the water stored in soil rock fragments can have an isotopic composition distinct to that of soil water or groundwater, being either relatively more depleted (in the case of $\delta^2H$ in Oshun et al., 2015), or more enriched (Palacio et al., 2014; Rong et al., 2011) "*

L90-91. It reads as if "the isotopic composition of rock water" is "a relevant, alternative plant water source". Please revise grammar.

**We have corrected the sentence as follows:**

*"Thus, wherever weathered rocks constitute a large fraction of the soil volume, the isotopic composition of rock moisture should be measured as rock moisture could constitute a significant alternative plant water source."*

L101. Do you mean "discrimination increased with decreasing soil water content"?

**No, we meant that "discrimination increased with soil water loss" (or cumulative transpiration), which is what the authors of that study reported. In their study, the explanatory variable was not "soil water content", but "soil water loss", estimated by gravimetric methods. For different soil textures or soil moisture levels, soil water loss could be equal but soil water content could differ significantly.**

L104-105. "In fact, a growing number of studies are observing lower xylem water isotopic compositions compared to those of the considered sources".

**We rephrased it to:**

**"In fact, a growing number of studies are reporting xylem water with an isotopic composition that is depleted relatively to that of the considered sources".**

L108. "The effect of deuterium fractionation on the quantification of sources' relative contributions to plant water uptake. . ."

**Changed.**

L109. "hydrogen isotopic composition only. . .".

**Changed to "hydrogen isotopes only"**

L114-121. This belongs to Material and Methods.

**We do not agree with the referee. This section describes the ecological questions addressed by our study. Even if the focus of the study is more related to issues with the application of stable isotope techniques, we are still interpreting the results from an ecological point of view in the Discussion. Because this section is not merely a site description, we considered preferable to keep it here at the end of the Introduction rather than moving it to the next section in the Material and Methods.**

L128. What do you mean by "extensive" here? Be more precise in the description of this objective already in the introduction.

**We meant that our dataset is rather large in comparison with similar studies (not all, obviously). We have reworded this sentence.**

*"In parallel to the ecological focus of our study, the reported isotopic dataset spanning a whole growing season was also used to explore the potential effect of isotopic fractionation on the quantification of tree water sources."*

L154. Of which two plots?

**Thank you for noting this. Non-dominant trees were selected in only 2 of the 3 sampling plots. We rephrased it to:**

*"in two of the plots".*

L155. "In order to measure the xylem water isotopic composition"

**Changed.**

L159-160. To "Each soil core was split into top soil (0-10 cm) and deep soil (from 70-80 to 110-120 cm depending on the depth of the rocky layer): even though you underline in the introduction the importance of identifying all water sources (i.e., hypothesis H2), you deliberately omit to sample the soil between 10 and 70(110) cm.

**The full answer to this comment is provided at the beginning of this letter.**

L178-179. I suggest to move this at the end of the §(L148).

**We moved this section as proposed, before "The studied area has a mean annual temperature…".**

L185. I propose: "the pressure in the extraction line. . ."

**Changed.**

L208. Which "significant difference"?

**It refers to the isotopic composition of xylem water and its sources. This is now clarified:**

*"Because no significant difference was found between the isotopic compositions of xylem (or water sources) between the different studied plots…"*

L209-214. A "mismatch" is not precise enough. You may write that you want to "assess if the isotopic compositions of the xylem water samples fall onto the evaporation line in a dual isotopic space". Also I would only present the concept of SW-Excess you are introducing and say that it is an adaptation of the LC-Excess.

**The evaporation line is not the correct concept here. Still, we replaced "mismatch" by "isotopic offset" in order to improve clarity.**

L211-212.To "However, because the source water for a tree is more likely to be made of soil water than rain water directly": it is a strange thing to say that trees would directly extract rain water just "less likely" than soil water.

**We agree that rain water can sometimes be the source of water of trees, especially just after summer rain events. However, in this case, this would also be reflected, at least partially, in the isotopic composition of topsoil water. Thus, the distinction between rain water use and soil water use seems adequate to us.**

L215. σ and Λ does not seem appropriate symbols, e.g., σ normally stands for standard error. I suggest other symbols like "c" and "d" or "A" and "B", "a'" and "b''etc.

**We have now substituted σ by Γ.**

L218. To "δ2H separation". Since by "δ2H", you mean a numerical value, you shouldn't use "separation", rather "offset", "difference" etc.

**Changed by "offset".**

L223-227. Why "models" (plural)? MixSIAR is one single model that has different scripts interacting with each other. To "Models were ran in the script version of the package": what other versions of MixSIAR are available? To "the number of Markov chain Monte-Carlo iterations was increased until convergence was reached": did you do this yourself, or was the number of runs optimized automatically? To "Gelman and Geweke diagnostics": what are these? To "residual error term in the isotope mixing models": what is this?

**We appreciate the comments of the referee, pointing out the lack of clarity in the description of our methodology. Although we only used MixSIAR, so the same model, every "run" is a differently parametrized mixing model. This is now clarified:**

**"The contribution of different water sources to that of xylem water was estimated using the MixSIAR package (Stock and Semmens, 2016) in R (R Core Development Team, 2012). Different mixing models were ran in the script version of the package…"**

**Yes, the number of MCMC iterations was not pre-set so we did it ourselves. Because the convergence can take more or less iterations to occur depending on the input data, we adjusted different number of iterations until we found that convergence was reached. This was checked for each individual mixing model. Gelman and Geweke are convergence diagnostics, so we based our decisions on the length of the chains based on a certain threshold value of these. This is now clarified:**

**"… and the number of Markov chain Monte-Carlo iterations was increased manually (by trial and error) until convergence was reached and the results for the Gelman and Geweke diagnostics were acceptable."**

**In MixSIAR, there is the possibility to calculate the source contribution for individual trees or for a group. In our case, we ran the models for all the trees in the same plot and from the same species. The authors of MixSIAR recommend to include the residual error term when calculating source contributions of grouped individuals, as it allows to account for unknown sources of error on the observed data. More details can be found in the MixSIAR package manual, or the related publications (Stock and Semmens, 2016, Parnell et al. 2010, see manuscript's reference list).**

L228. Please define "top" and "deep soil water".

**This is already defined in section 2.1.**

L231-241. You only test the sensitivity of the isotopic mixing model to the definition of the "product", meaning the sap xylem isotopic composition. What about the definition of the "sources"? I am afraid that, since you do not fully characterize the ensemble of potential water sources, you cannot assess their contributions to tree RWU (see my general comment).

**This point is discussed in the response to the general comment of the referee.**

L257. To "3.2. Stable isotopes of tree water sources": do the tree water sources have distinct "stable isotopes" than other sources:)? You might write something like "Stable isotopic composition distribution across tree water sources". Same goes for 3.3.

**We have modified this according to the suggestion of the referee.**

L258-259. Belongs to M&M.

**We have moved this sentence to section 2.1 of Material and Methods and added a shorter sentence here:**

**"The long-term (2007-present) local meteoric water line (LMWL) using the closest GNIP station (see Material and Methods) is shown in each panel of Fig. 2."**

L266. A slope value of 9.99? Is this a typo? If not, what is the p-value of the linear regression?

**No, this is not a typo. The sampling of July 4$^{th}$ was done following a 5-day rain episode that left more than 100 mm of rain. This caused that the top soil water became more depleted than the deep soil water, producing a soil water line with a slope steeper than that of the LMWL. The p-value of the regression was <0.0001. This can be noted in the corresponding panel of Fig. 2. But note also that the soil water line on this date is clearly not an evaporation line (otherwise topsoil should be more enriched than deep soil), i.e., the slope should be reported as -10 rather than +10…**

L275. "in both hydrogen and oxygen heavier isotopes"

**We have reworded this sentence:**

**"Finally, $\delta^2H$ and $\delta^{18}O$ of rock moisture were significantly more enriched than those of top and deep soil water"**

L277. "The isotopic composition of rock moisture". In general, I suggest that you avoid use of "signal" throughout the MS. L281. No comma in ". . .budburst), was. . ."

**We consider that the term isotopic signal used as a synonym of isotopic composition is widely accepted and understood by the community. In some cases and with the aim of improving readability, we consider it as the most appropriated term. However, we would change it all over the manuscript if the editor considers that it would improve the text or its comprehension.**

L284. "had a lower δ2H"

**In our opinion, stating that the isotopic composition of X is more depleted/enriched than that of Y is an appropriate expression. It would not be correct just using it to characterise a single water, but we consider it correct when used in relative terms to compare between two or more waters.**

L289. "F. sylvatica presented higher δ18O values (P < 0.05)". Same goes for the rest of the MS: a value cannot be enriched/depleted.

**Same as above.**

L305-333. You are using the same Bayesian inference model with different parameterization, not a series of different mixing models. This might be confusing to the readers.

**We find it appropriate to speak about different models, even if the type of model is the same. We believe that this use of the plural of models is commonly accepted and understood. For instance, when running Generalized Linear Mixed (GLM) models, one can speak about different models when using**

different response and predictor variables, all they are all GLM models. If the editor thinks that our analyses can be confused by the readers with a comparison between isotopic mixing models (IsoSource, SIAR, MixSIR, MixSIAR), we could modify the text accordingly.

L338. "to identify plant water sources and quantify their relative contributions to tree RWU. . ."

**We thank the referee for this suggestion. We have modified the beginning of the Discussion (see the following response).**

L338-341. I propose something like "lower xylem water isotopic composition than any combination of those of the identified water sources would give".

**We have modified the sentence as follows:**

*"Our results support those from recent studies reporting xylem water with an hydrogen isotope composition more depleted than any potential water source, and thus of any of their combination."*

L341-342. To "The diversity of methodologies used for the extraction of waters and their isotopic determination in all these studies, including ours, rules out potential analytical bias.". Why could these methods not all be biased? Orlowski et al. (2016a) and Orlowski et al. (2016b) have shown discrepancies between the different methods for water extraction.

**Here we meant that similar results have been reported using all sorts of water extraction and isotopic determination techniques. Issues associated with these techniques may produce a diversity of biases, e.g. extracted soil water being more depleted than input water (bulk soil), positive or negative effects of different types of clay-related cations on soil water, or organic interference during isotopic determination producing enriched/depleted isotopic compositions respective to the original sample. These effects can be of opposed sign, which minimizes the likelihood of a by chance commonly observed phenomenon. In addition, in our case, the results could not be explained by any of these biases, at least those that have been already reported in the literature. Still, we acknowledge that minimizing the likelihood is not the same as completely ruling out. We have thus modified this sentence:**

*"The diversity of methodologies used for the extraction of waters and their isotopic determination in all these studies, including ours, minimises the likelihood of a common analytical or methodological bias"*

**In addition, we have now added a full paragraph discussing the possibility of fractionation processes occurring during cryogenic extraction for our particular case. Notably, we argue that reported effects effects of carbonates in the soil water isotopic composition could not be responsible for the isotopic offset observed here and thus would modify the conclusion drawn.**

*"Another possibility is that fractionation processes occur during water extraction. Meißner et al., (2014) reported that treating soil samples with HCl to remove carbonates prior to water extraction led to a cryogenically-extracted water $\delta^{18}O$ in agreement with that of input water, whereas the $\delta^{18}O$ of cryogenically-extracted water from carbonate-rich soil samples was depleted by about 1‰ compared to input water. On the other hand, they found no effect of carbonate content on hydrogen isotopes. They suggested that the $\delta^{18}O$ depletion of extracted water was caused by oxygen isotope exchanges between soil water and carbonates during the extraction, a process that should be temperature-dependent. Meißner et al. (2014) did not specify their extraction temperature but we expect it to be > 60°C, i.e. close to our extraction temperature of 80°C, so that we could expect a carbonate-induced*

*isotope effect of comparable magnitude. If the presence of carbonates in the C horizon were responsible for a $\delta^{18}O$ depletion of extracted water from the deep soil samples of about 1‰, this would mean that the "true" soil water in this horizon should be shifted by about +1‰. This would slightly modify the SW-excess values but would not cancel the observed isotopic offset between soil water and xylem water. Therefore, although the results of Meißner et al. (2014) are very relevant to our study, they cannot explain the isotopic offset observed here"*

L343 and after. If you mean by "offset" the "SW-Excess", then say so. Otherwise, define each time what are these offsets (e.g., is offset the same as in the study of Evaristo et al., 2017?)

**We have now specified that we refer here to "$\delta^2H$" offsets between xylem and source waters (see above). The SW-excess is an approach to quantify the isotopic offset between xylem water and soil water but we wanted to also include offsets with any potential source or their combinations.**

L344. "Sternberg"

**Changed. We also rephrased this sentence:**

**"Furthermore, isotopic offsets between soil and xylem water in potted plants (Ellsworth & Stenberg 2007; Vargas *et al.*, 2017) and botanical gardens (Evaristo et al., 2017) have also been reported and discussed these to some extent."**

L380. How do you define a "similar offset" for δ2H and δ18O? Stating that both are linearly linked (e.g., δ2H = a*δ18O + b), would it mean that Δδ2H is similar as Δδ18O if Δδ2H = a*Δδ18O ??

**We have completely rewritten this section:**

**"Thus, although empirical evidence for an isotope separation between bulk and plant-accessed soil water pools is growing, we do not think this is the cause of the isotopic offset reported here. Otherwise, we would expect both hydrogen and oxygen isotopes to be affected and the isotope separation between plant and bulk soil waters to be weaker when soil water content is large. Instead we found for all our trees a significant $\delta^2H$ offset between xylem and soil water sources (Fig. 5), even at times when soil water content was large (Fig. 1).**

**Similarly we de not think that branch evaporation is responsible for the reported isotopic offset (Martín-Gómez et al., 2017). If it were the case, we would expect to vary over the season with evaporative demand and to affect both hydrogen and oxygen isotopes, i.e., the opposite of what is found here.**

**We found differences in SW-excess when xylem water was collected from coarse roots rather than twigs (Fig. 4). Previous studies have shown that water in coarse or tap roots can exhibit significant depletion in $\delta^2H$ relative to source water pools for examples in *Populus euphratica* (Zhao et al., 2016) and *Prosopis velutina* (Ellsworth and Williams, 2007). Moreover, the $\delta^2H$ offsets reported between soil and root water were of the same order of magnitude (ca. -20‰ for *P. euphratica* and ca. -7‰ for *P. velutina*) as observed here for *F. sylvatica* and *Q. robur* (Fig. 4 and Fig. 5). Interestingly, Zhao *et al.*, (2016) analysed xylem water and what they called tissue water (*i.e.* bulk plant water) separately with the former not showing any isotopic depletion compared to source water. In this context, Ellsworth & Williams, (2007) attributed…"**

L416. Rothfuss and Javaux (2016) is the discussion paper, not the actual article (Rothfuss and Javaux, 2017), please edit this.

**We have corrected this throughout the manuscript.**

L419. Define "SWexcessx" (i.e., already in the M&M).

**This was a remaining from a previous draft. We are now using the term SW-excess throughout the manuscript.**

L422-424. Using both δ18O and δ2H values does not mean you work in a bidimensional space. This is strictly speaking a 2x1D space since δ18O and δ2H across water sources are linearly linked. To be able to assess relative contributions to RWU in a 2D space, you need to break this linear bond (via isotopic labeling; e.g., Beyer et al., 2016), so that these water sources spread on some area rather than on some line in a (δ18H, δ2H) coordinate system.

**We disagree because the fractionation factors during natural processes do not affect the two water isotopes similarly, i.e. they are not related in a mass-dependent fashion as for $\delta^{17}$O and $\delta^{18}$O for example. In any case, we do not understand how this comment relates to the lines 422-424.**

L427. "deuterium fractionation"

**Changed to "hydrogen isotope fractionation".**

**References**

Beyer, M., Koeniger, P., Gaj, M., Hamutoko, J. T., Wanke, H., and Himmelsbach, T.: A deuterium-based labeling technique for the investigation of rooting depths, water uptake dynamics and unsaturated zone water transport in semiarid environments, J. Hydrol., 533, 627-643, 10.1016/j.jhydrol.2015.12.037, 2016.

Gaj, M., Beyer, M., Koeniger, P., Wanke, H., Hamutoko, J., and Himmelsbach, T.: In-situ unsaturated zone stable water isotope (2H and 18O) measurements in semi-arid environments using tunable off-axis integrated cavity output spectroscopy, hydrol. Earth Syst. Sci., 20, 715-731, 10.5194/hess-20-715-2016, 2015.

Oerter, E. J., Perelet, A., Pardyjak, E., and Bowen, G.: Membrane inlet laser spectroscopy to measure H and O stable isotope compositions of soil and sediment pore water with high sample throughput, Rapid Commun Mass Spectrom, 10.1002/rcm.7768, 2016.

Orlowski, N., Breuer, L., and McDonnell, J. J.: Critical issues with cryogenic extraction of soil water for stable isotope analysis, Ecohydrology, 9, 3-10, 10.1002/eco.1722, 2016a.

Orlowski, N., Pratt, D. L., and McDonnell, J. J.: Intercomparison of soil pore water extraction methods for stable isotope analysis, Hydrol. Process., 30, 3434-3449, 10.1002/hyp.10870, 2016b.

Rothfuss, Y., Vereecken, H., and Brüggemann, N.: Monitoring water stable isotopic composition in soils using gas-permeable tubing and infrared laser absorption spectroscopy, Water Resour. Res., 49, 1-9, 10.1002/wrcr.20311, 2013.

Rothfuss, Y., and Javaux, M.: Reviews and syntheses: Isotopic approaches to quantify root water uptake: a review and comparison of methods, Biogeosciences, 14, 2199- 2224, 10.5194/bg-14-2199-2017, 2017.

Volkmann, T. H., Kühnhammer, K., Herbstritt, B., Gessler, A., and Weiler, M.: A method for in situ monitoring of the isotope composition of tree xylem water using laser spectroscopy, Plant Cell Environ, 10.1111/pce.12725, 2016.

Volkmann, T. H. M., and Weiler, M.: Continual in situ monitoring of pore water stable isotopes in the subsurface, Hydrol. Earth Syst. Sc., 18, 1819-1833, 10.5194/hess-18- 1819-2014, 2014.